# Multi-contrast anatomical subcortical structures parcellation

Pierre-Louis Bazin[1,2]*, Anneke Alkemade[1], Martijn J Mulder[1,3], Amanda G Henry[4], Birte U Forstmann[1]

[1]Integrative Model-based Cognitive Neuroscience research unit, University of Amsterdam, Amsterdam, Netherlands; [2]Max-Planck Institute for Human Cognitive and Brain Sciences, Leipzig, Germany; [3]Psychology Department, Utrecht University, Utrecht, Netherlands; [4]Faculty of Archaeology, Leiden University, Leiden, Netherlands

**Abstract** The human subcortex is comprised of more than 450 individual nuclei which lie deep in the brain. Due to their small size and close proximity, up until now only 7% have been depicted in standard MRI atlases. Thus, the human subcortex can largely be considered as terra incognita. Here, we present a new open-source parcellation algorithm to automatically map the subcortex. The new algorithm has been tested on 17 prominent subcortical structures based on a large quantitative MRI dataset at 7 Tesla. It has been carefully validated against expert human raters and previous methods, and can easily be extended to other subcortical structures and applied to any quantitative MRI dataset. In sum, we hope this novel parcellation algorithm will facilitate functional and structural neuroimaging research into small subcortical nuclei and help to chart terra incognita.

*For correspondence:
pilou.bazin@uva.nl

Competing interests: The authors declare that no competing interests exist.

## Introduction

Subcortical brain structures are often neglected in neuroimaging studies due to their small size, limited inter-regional contrast, and weak signal-to-noise ratio in functional imaging (*Forstmann et al., 2016*; *Johansen-Berg, 2013*). Yet, these small and diverse structures are prominent nodes in functional networks (*Marquand et al., 2017*; *Ji et al., 2019*), and they undergo pathological alterations already at early stages of neurodegenerative diseases (*Andersen et al., 2014*; *Koshiyama et al., 2018*). Deep brain stimulation surgery, originally performed to reduce motor symptoms in essential tremors, is now a promising therapeutic option in later stages of Parkinson's disease and movement disorders, as well as refractory psychiatric illnesses in obsessive-compulsive disorder, anorexia, or depression (*Forstmann et al., 2017*; *Mosley et al., 2018*). Evolutionary genetics even uncovered that in modern humans, Neanderthal-inherited alleles were preferentially down-regulated in subcortical and cerebellar regions compared to other brain regions (*McCoy et al., 2017*), suggesting these structures to be essential in making us specifically human.

Despite their importance, these areas are particularly difficult to image. Furthermore, the size, shape, and location of these brain regions changes with development and aging (*Fjell et al., 2013*; *Keuken et al., 2013*; *Yeatman et al., 2014*; *Herting et al., 2018*). Experience-based plasticity continuously remodels myelin (*Tardif et al., 2016*; *Hill et al., 2018*; *Turner, 2019*), iron and other magnetic substances accumulate with age or pathology (*Andersen et al., 2014*; *Zhang et al., 2018*), both bringing changes in the MRI appearance of subcortical regions with diverse tissue characteristics (*Draganski et al., 2011*; *Keuken et al., 2017*).

Thus, mapping the structure and function of the subcortex is a major endeavor as well as a major challenge for human neuroscience. Extensive work available from animal brain models unfortunately does not translate in a straightforward way to human subcortical anatomy nor does it shed much light on its involvement in human cognition (*Steiner and Tseng, 2017*). Besides serious difficulties in

obtaining adequate measures of subcortical neural activity in functional MRI (*de Hollander et al., 2017*; *Miletić et al., 2020*), atlases and techniques for labeling accurately and reliably individual subcortical structures have also been scarce (*Frazier et al., 2005*; *Chakravarty et al., 2006*; *Ahsan et al., 2007*; *Yelnik et al., 2007*; *Qiu et al., 2010*; *Patenaude et al., 2011*), typically labeling the thalamus, striatum (or its subdivision into caudate and putamen), and globus pallidus (internal and external segments combined), sometimes the amygdala. However, recent advances in anatomical MRI, combining multiple contrasts and/or quantitative MRI mapping and utilizing the higher resolution achievable with 7 Tesla (7T) and above have started to reduce the gap, each mapping a few additional structures or sub-structures, primarily the iron-rich substantia nigra, red nucleus and subthalamic nucleus (*Keuken et al., 2013*; *Xiao et al., 2015*; *Visser et al., 2016a*; *Visser et al., 2016b*; *Wang et al., 2016*; *Makowski et al., 2018*; *Ewert et al., 2018*; *Iglesias et al., 2018*; *Pauli et al., 2018*; *Sitek et al., 2019*). While these efforts generated valuable atlases, they do not yet enable to identify many subcortical structures in individual subjects. Manual delineation, on the other hand, requires extensive manual labor from highly trained experts which cannot be easily applied to large cohorts or clinical settings.

Here, we propose a new automated parcellation technique to identify and label 17 individual subcortical structures of varying size and composition in individual subjects, based on a large quantitative 7T MRI database (*Alkemade et al., 2020*), using quantitative maps of relaxation rates R1 and R2* (1/T1 and 1/T2*, respectively) and quantitative susceptibility maps (QSM) as anatomical contrasts. The algorithm, named Multi-contrast Anatomical Subcortical Structure Parcellation (MASSP), follows a Bayesian multi-object approach similar in essence to previous efforts (*Fischl et al., 2002*; *Eugenio Iglesias et al., 2013*; *Visser et al., 2016a*; *Garzón et al., 2018*), combining shape priors, intensity distribution models, spatial relationships, and global constraints. The main innovation of our approach is to explicitly estimate interfaces between subcortical structures based on a joint model derived from signed distance functions. Modeling interfaces in addition to the structure itself provides a rich basis to encode relationships and anatomical knowledge in shape and intensity priors. A voxel-wise Markovian diffusion regularizes the combined priors for each defined interface, lowering the imaging noise. Finally, the voxel-wise posteriors for the different structures and interfaces are further combined into global anatomical parcels by topology correction and region growing taking into account volumetric priors, which regularizes parcellation results further in smaller nuclei with low or heterogeneous contrast. To validate the results from this new method, in a thorough comparison with expert manual labeling, we show that the proposed method provides results very close from manual raters in many structures and exhibit reasonable bias across the adult lifespan. The method can easily be extended to new structures, can be applied to any quantitative MRI dataset and is available in Open Source as part of Nighres (*Huntenburg et al., 2018*), a neuroimage analysis package aimed at high-resolution neuroimaging.

## Results

The MASSP parcellation method presented here has been trained to parcellate the following 17 structures: striatum (Str), thalamus (Tha), lateral, 3rd and 4th ventricles (LV, 3V, 4V), amygdala (Amg), globus pallidus internal segment (GPi) and external segment (GPe), SN, STN, red nucleus (RN), ventral tegmental area (VTA), fornix (fx), internal capsule (ic), periaqueductal gray (PAG), pedunculopontine nucleus (PPN), and claustrum (Cl), see *Figure 1*. These structures include the most commonly defined subcortical regions (Str, Tha, Amg, LV), the main iron-rich nuclei (GPi, GPe, RN, SN, STN), as well as smaller, less studied areas (VTA, PAG, PPN, Cl), white matter structures (ic, fx), and the central ventricles (3V, 4V).

MASSP uses a data set of ten expert delineations as a basis for its modeling. From the delineations, an atlas of interfaces between structures, shape skeletons, and interface intensity histograms are generated, and used as prior in a multiple-step non-iterative Bayesian algorithm, see *Figure 2* and Materials and methods.

### Validation against manual delineations

In a leave-one-out validation study comparing performance with the manual delineations, MASSP performed above 95% of the level of quality of the raters for Str, Tha, 4V, GPe, SN, RN, VTA, ic in terms of Dice overlap, the most stringent of the quality measures (see *Figures 3* and *4* and *Table 1*).

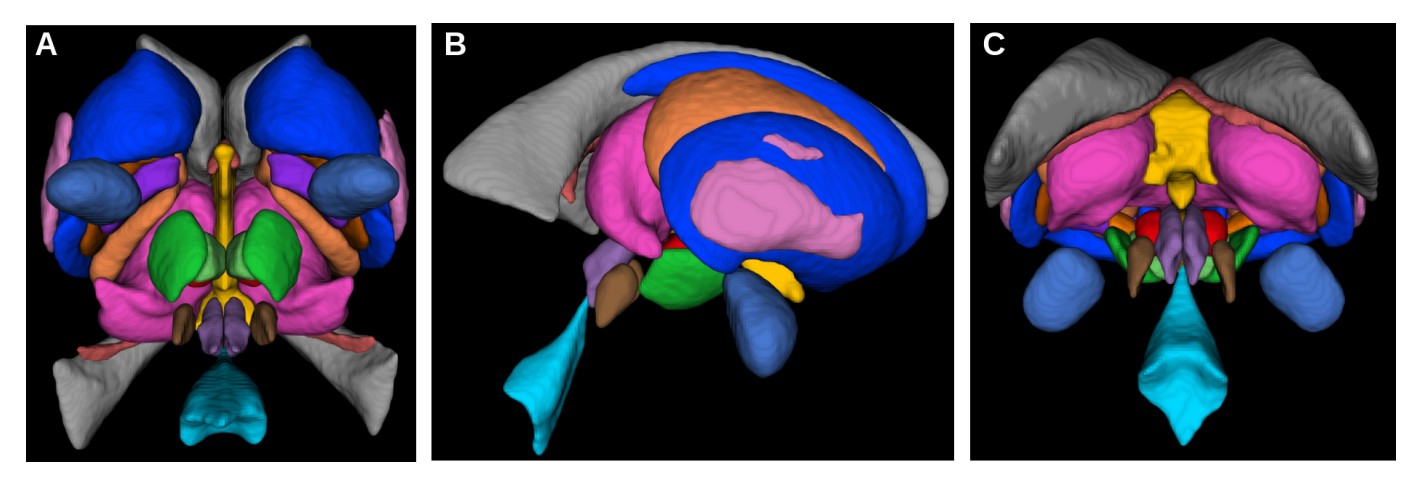

**Figure 1.** The 17 subcortical structures currently included in the parcellation algorithm in axial (**A**), sagittal (**B**), and coronal (**C**) views.

Several of the smaller structures have lower overlap ratios likely due to their smaller size (GPi, STN, PAG, PPN). Structures with an elongated shape (fx, Cl) remain challenging, due to the fact that small differences in location can substantially reduce overlap (*Bazin et al., 2016*). Despite these challenges, when comparing the dilated Dice scores, all structures were above 75% of overlap, with most reaching over 90% of the manual raters ability. Note that the Dice coefficient is very sensitive to size, as smaller structures will have lower overlap ratios for the same number of misclassified voxels. The dilated Dice coefficient is more representative of the variability regardless of size, as the smaller structures can reach high levels of overlap, both in manual and automated parcellations (see *Table 1*). The average surface distance confirms these results, showing values generally between one and two voxels of distance at a resolution of 0.7 mm, except in the cases of Amg, LV, fx, PPN, and Cl. These structures are generally more variable (LV), elongated (fx, Cl), or have a particularly low contrast with neighboring regions (Amg, PPN).

## Comparison to other automated methods

To provide a basis for comparison, we applied other freely available methods for subcortical structure parcellation to the same 10 subjects. MASSP performs similarly to or better than Freesurfer, FSL FIRST and a multi-atlas registration using ANTs (see *Table 2*). Multi-atlas registration provides high accuracy in most structures as well, but is biased toward under-estimating the size of smaller and elongated structures where overlap is systematically reduced across the individual atlas subjects. Multi-atlas registration is also quite computationally intensive when using multiple contrasts at high

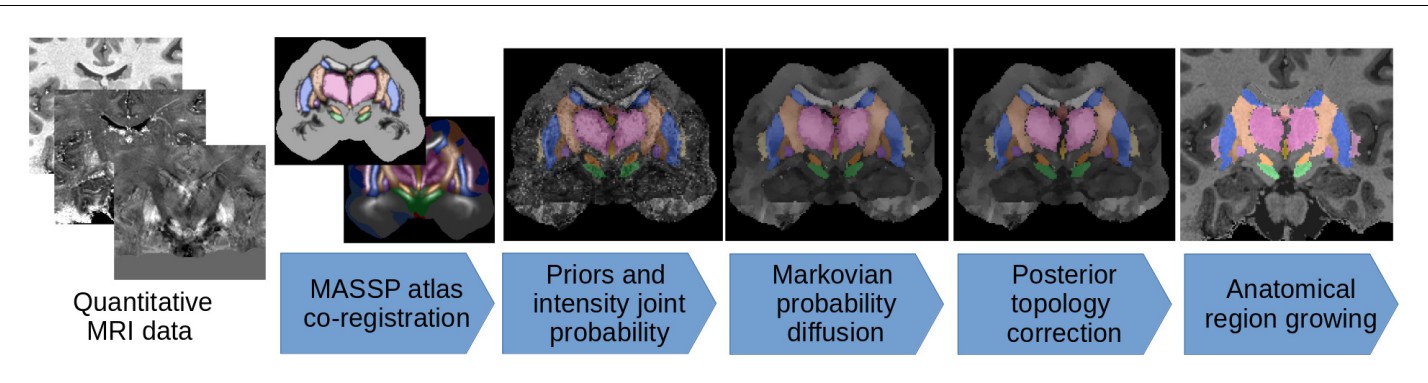

**Figure 2.** The MASSP parcellation pipeline. Atlas priors for interfaces between structures are combined to the MRI data, regularized via probability diffusion and topology correction, and the final structure posteriors are jointly estimated by region growing.

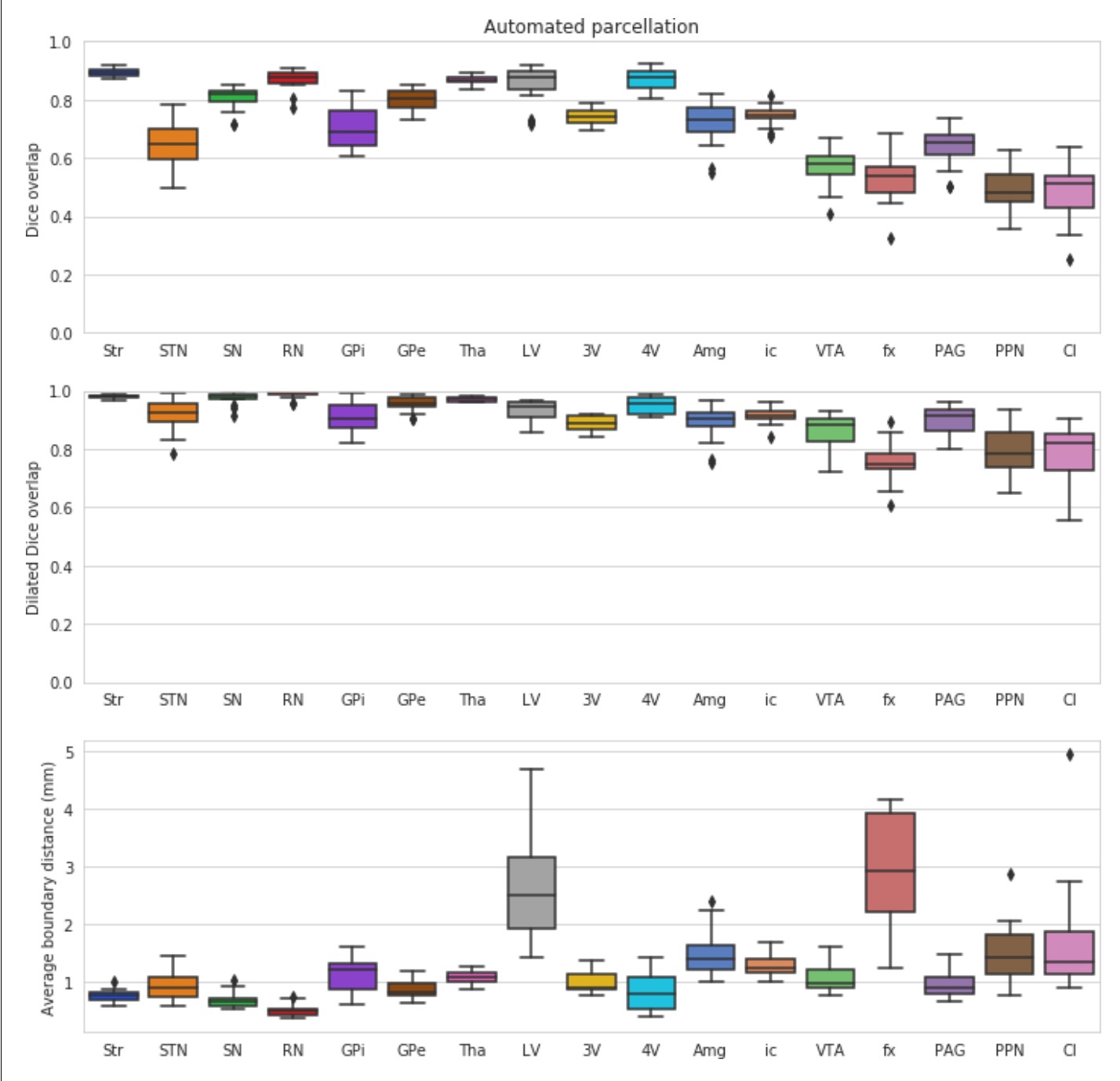

**Figure 3.** Leave-one-out validation of the structures parcellated by MASSP, compared to the human rater with most neuroanatomical expertise. Scores for the left and right side are computed separately and then combined into box-and-whisker plots.

resolution. Finally, MASSP provides many more structures than Freesurfer and FSL FIRST, and can be easily applied to new structures based on additional manual delineations.

## Application to new MRI contrasts

Quantitative MRI has only become recently applicable in larger studies, thanks in part to the development of integrated multi-parameter sequences (*Weiskopf et al., 2013*; *Caan et al., 2019*). Many data sets, including large-scale open databases, use more common T1- and T2-weighted MRI. In

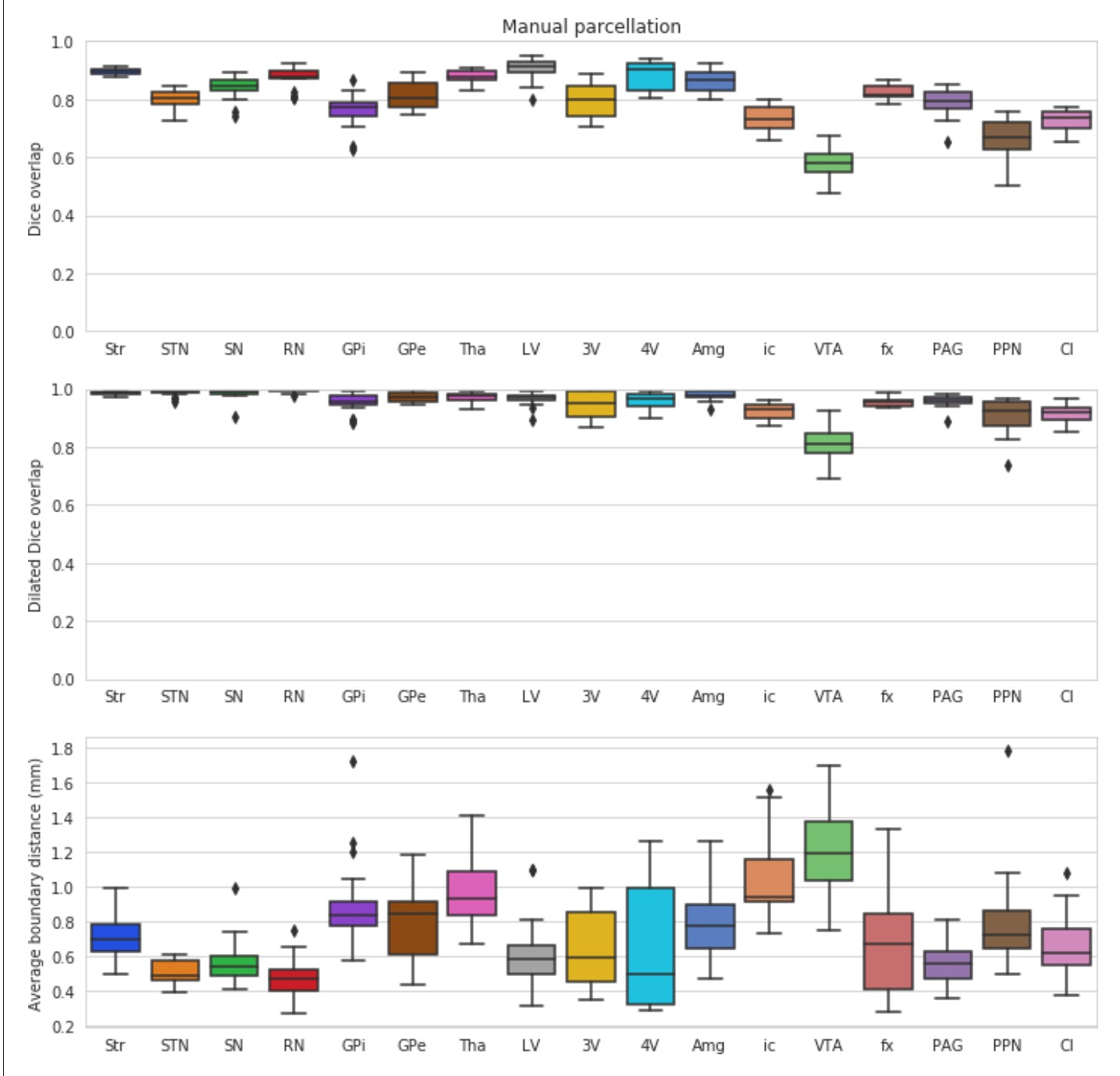

**Figure 4.** Inter-rater variability for the human expert raters.  Scores for the left and right side are computed separately and then combined into box-and-whisker plots.

order to test the applicability of MASSP to such contrasts, we obtained the test-retest subset of the Human Connectome Project (HCP, *Van Essen et al., 2013*) and applied MASSP to the 45 pre-processed and skull-stripped T1- and T2-weighted images from each of the two test and retest sessions. While performing manual delineations on the new contrasts would be preferable, the model is already rich enough to provide stable parcellations. Test-retest reproducibility is similarly high for MASSP and Freesurfer, and are generally in agreement, see *Figure 5* and *Table 3*.

**Table 1.** Mean overlap and distance measures for the leave-one-out validation.

| | Str | STN | SN | RN | GPi | GPe | Tha | LV | 3V | 4V | Amg | ic | VTA | fx | PAG | PPN | Cl |
|---|---|---|---|---|---|---|---|---|---|---|---|---|---|---|---|---|---|
| | | | | | | | **Dice overlap** | | | | | | | | | | |
| MASSP | 0.893 | 0.648 | 0.805 | 0.870 | 0.702 | 0.800 | 0.867 | 0.849 | 0.741 | 0.869 | 0.723 | 0.745 | 0.570 | 0.527 | 0.641 | 0.496 | 0.485 |
| Manual | 0.897 | 0.800 | 0.841 | 0.875 | 0.762 | 0.813 | 0.877 | 0.907 | 0.797 | 0.882 | 0.866 | 0.732 | 0.574 | 0.823 | 0.791 | 0.665 | 0.727 |
| Ratio | 0.995 | 0.811 | 0.957 | 0.996 | 0.925 | 0.987 | 0.989 | 0.936 | 0.936 | 0.988 | 0.836 | 1.020 | 0.994 | 0.641 | 0.814 | 0.754 | 0.664 |
| | | | | | | | **Dilated overlap** | | | | | | | | | | |
| MASSP | 0.982 | 0.919 | 0.977 | 0.991 | 0.909 | 0.956 | 0.970 | 0.929 | 0.890 | 0.951 | 0.891 | 0.915 | 0.863 | 0.756 | 0.897 | 0.795 | 0.789 |
| Manual | 0.987 | 0.988 | 0.985 | 0.995 | 0.953 | 0.972 | 0.970 | 0.967 | 0.944 | 0.961 | 0.978 | 0.924 | 0.818 | 0.957 | 0.960 | 0.910 | 0.914 |
| Ratio | 0.995 | 0.930 | 0.992 | 0.995 | 0.955 | 0.984 | 1.000 | 0.961 | 0.946 | 0.991 | 0.911 | 0.992 | 1.059 | 0.790 | 0.935 | 0.879 | 0.863 |
| | | | | | | | **Average surface distance** | | | | | | | | | | |
| MASSP | 0.750 | 0.911 | 0.676 | 0.491 | 1.140 | 0.863 | 1.058 | 2.690 | 0.994 | 0.817 | 1.476 | 1.275 | 1.074 | 2.950 | 0.955 | 1.484 | 1.685 |
| Manual | 0.723 | 0.508 | 0.571 | 0.482 | 0.902 | 0.804 | 0.971 | 0.615 | 0.637 | 0.671 | 0.779 | 1.045 | 1.204 | 0.703 | 0.555 | 0.801 | 0.670 |
| Ratio | 0.971 | 0.590 | 0.852 | 0.996 | 0.861 | 0.943 | 0.916 | 0.277 | 0.662 | 1.020 | 0.553 | 0.834 | 1.161 | 0.287 | 0.610 | 0.619 | 0.465 |

## Biases due to atlas size

A common concern of brain parcellation methods is the risk of biases, as they are typically built from a small number of manual delineations. Our data set is part of a large scale study of the subcortex, for which we obtained manual delineations of the STN, SN, RN, GPe, and GPi on 105 subjects over the adult lifespan (18–80 year old, see *Alkemade et al., 2020* for details). First, we investigated the impact of atlas size. We randomly assigned half of the subjects from each decade to two groups, and built atlas priors from subsets of 3, 5, 8, 10, 12, 15, and 18 subjects from the first group. The subjects used in the atlas were taken randomly from each decade (18-30, 31-40, 41-50,51-60, 61-70, 71-80), so as to maximize the age range represented in each atlas. Atlases of increasing size were

**Table 2.** Comparison with multi-atlas registration, Freesurfer, and FSL FIRST.

| | Str | STN | SN | RN | GPi | GPe | Tha | LV | 3V | 4V | Amg | Ic | VTA | Fx | PAG | PPN | Cl |
|---|---|---|---|---|---|---|---|---|---|---|---|---|---|---|---|---|---|
| | | | | | | | **Dice overlap** | | | | | | | | | | |
| MASSP | 0.893 | 0.648 | 0.805 | 0.870 | 0.702 | 0.800 | 0.867 | 0.849 | 0.741 | 0.869 | 0.723 | 0.745 | 0.570 | 0.527 | 0.641 | 0.496 | 0.485 |
| Multi-atlas | 0.855 | 0.662 | 0.760 | 0.820 | 0.742 | 0.796 | 0.859 | 0.734 | 0.660 | 0.691 | 0.761 | 0.718 | 0.626 | 0.478 | 0.674 | 0.539 | 0.398 |
| Freesurfer | 0.876 | | | | 0.778 | | 0.838 | 0.858 | 0.430 | 0.769 | 0.692 | | | | | | |
| FSL FIRST | 0.875 | | | | 0.813 | | 0.839 | | | | 0.653 | | | | | | |
| | | | | | | | **Dilated overlap** | | | | | | | | | | |
| MASSP | 0.982 | 0.919 | 0.977 | 0.991 | 0.909 | 0.956 | 0.970 | 0.929 | 0.890 | 0.951 | 0.891 | 0.915 | 0.863 | 0.756 | 0.897 | 0.795 | 0.789 |
| Multi-atlas | 0.976 | 0.938 | 0.968 | 0.989 | 0.947 | 0.968 | 0.970 | 0.920 | 0.920 | 0.908 | 0.921 | 0.939 | 0.924 | 0.798 | 0.943 | 0.871 | 0.811 |
| Freesurfer | 0.975 | | | | 0.922 | | 0.946 | 0.974 | 0.562 | 0.911 | 0.857 | | | | | | |
| FSL FIRST | 0.976 | | | | 0.946 | | 0.950 | | | | 0.843 | | | | | | |
| | | | | | | | **Average surface distance** | | | | | | | | | | |
| MASSP | 0.750 | 0.911 | 0.676 | 0.491 | 1.140 | 0.863 | 1.058 | 2.690 | 0.994 | 0.817 | 1.476 | 1.275 | 1.074 | 2.950 | 0.955 | 1.484 | 1.685 |
| Multi-atlas | 0.961 | 0.891 | 0.858 | 0.675 | 0.992 | 0.882 | 1.083 | 1.417 | 0.932 | 1.249 | 1.359 | 1.129 | 0.813 | 1.362 | 0.794 | 1.055 | 1.273 |
| Freesurfer | 0.770 | | | | 1.211 | | 1.405 | 0.685 | 4.071 | 1.361 | 1.749 | | | | | | |
| FSL FIRST | 0.867 | | | | 1.143 | | 1.675 | | | | 1.746 | | | | | | |
| | | | | | | | **Volume bias** | | | | | | | | | | |
| MASSP | 0.041 | 0.017 | -0.038 | 0.007 | 0.066 | 0.089 | 0.040 | 0.0470 | 0.121 | 0.047 | 0.078 | 0.183 | 0.032 | -0.016 | 0.026 | 0.009 | 0.023 |
| Multi-atlas | 0.020 | -0.087 | -0.009 | 0.031 | 0.009 | 0.014 | 0.020 | -0.003 | -0.007 | -0.092 | -0.038 | 0.055 | -0.067 | -0.264 | -0.090 | -0.269 | -0.376 |
| Freesurfer | 0.017 | | | | 0.087 | | 0.163 | 0.122 | -0.551 | 0.351 | 0.468 | | | | | | |
| FSL FIRST | -0.100 | | | | -0.021 | | 0.165 | | | | -0.249 | | | | | | |

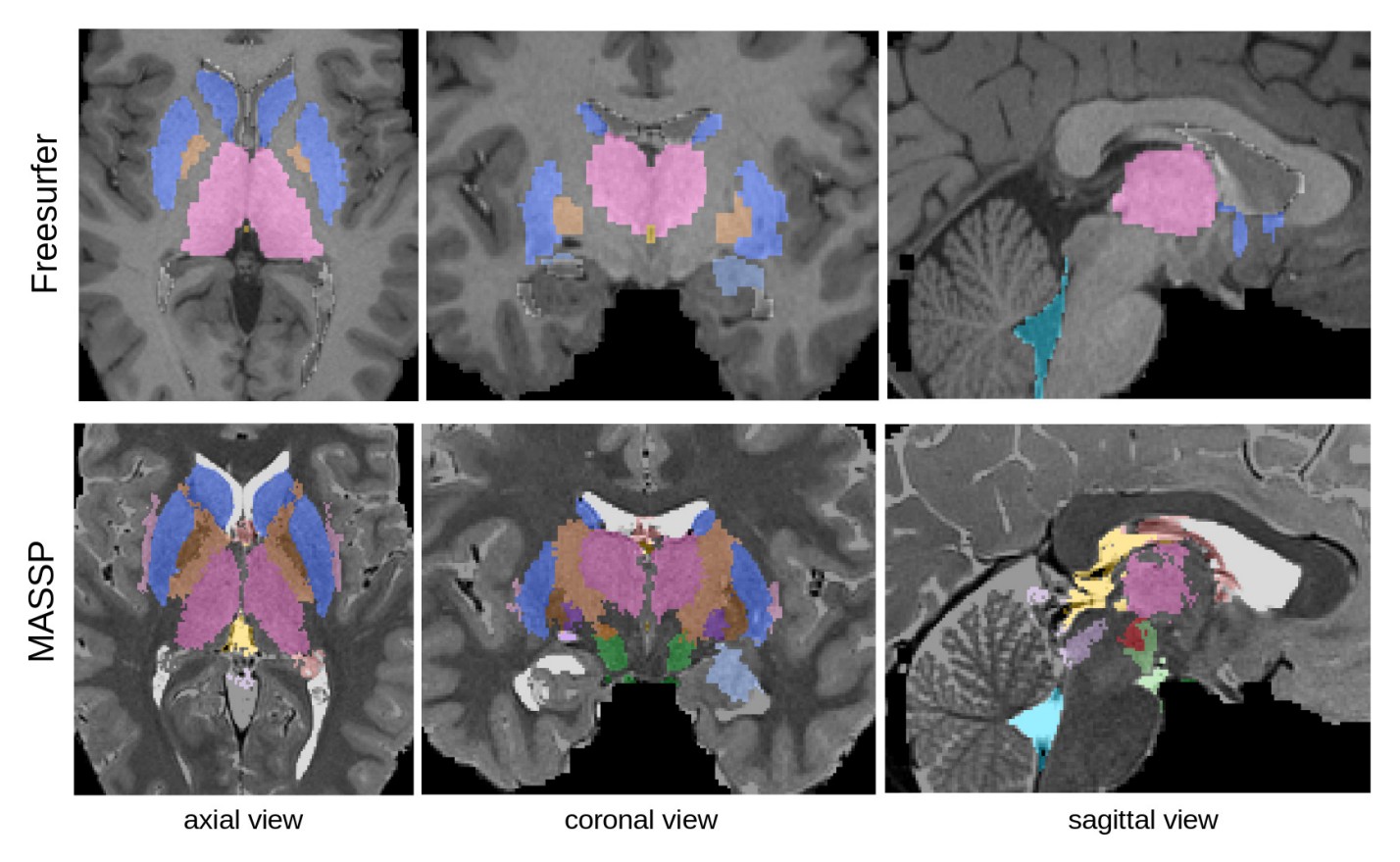

**Figure 5.** Parcellation with Freesurfer (top, on T1w image) and MASSP (bottom, on T2w image) on Human Connectome Project data. MASSP priors were not derived from the contrasts, but transferred via a spatial mapping of the quantitative MRI intensities from AHEAD subjects.

constructed by adding subjects to previous atlases, so that atlases of increasing complexity include all subjects from simpler atlases. Results applying these atlases to parcellate the second group are given in *Figure 6*. As in previous studies (*Eugenio Iglesias et al., 2013*; *Bazin and Pham, 2008*), performance quickly stabilized with atlases of more than five subjects (no significant difference in

**Table 3.** Test-retest comparison with Freesurfer on Human Connectome Project data.

| | Str | STN | SN | RN | GPi | GPe | Tha | LV | 3V | 4V | Amg | ic | VTA | fx | PAG | PPN | Cl |
|---|---|---|---|---|---|---|---|---|---|---|---|---|---|---|---|---|---|
| **Dice overlap** | | | | | | | | | | | | | | | | | |
| MASSP test-retest | 0.914 | 0.701 | 0.818 | 0.829 | 0.791 | 0.859 | 0.928 | 0.881 | 0.837 | 0.870 | 0.866 | 0.860 | 0.738 | 0.774 | 0.714 | 0.713 | 0.785 |
| Freesurfer test-retest | 0.898 | | | | 0.770 | | 0.919 | 0.894 | 0.842 | 0.849 | 0.852 | | | | | | |
| MASSP – Freesurfer | 0.876 | | | | 0.778 | | 0.838 | 0.858 | 0.430 | 0.769 | 0.692 | | | | | | |
| **Dilated overlap** | | | | | | | | | | | | | | | | | |
| MASSP test-retest | 0.987 | 0.939 | 0.977 | 0.978 | 0.963 | 0.977 | 0.990 | 0.980 | 0.979 | 0.986 | 0.981 | 0.973 | 0.969 | 0.961 | 0.965 | 0.972 | 0.966 |
| Freesurfer test-retest | 0.986 | | | | 0.926 | | 0.986 | 0.989 | 0.972 | 0.975 | 0.978 | | | | | | |
| MASSP – Freesurfer | 0.954 | | | | 0.788 | | 0.919 | 0.934 | 0.435 | 0.901 | 0.866 | | | | | | |
| **Average surface distance** | | | | | | | | | | | | | | | | | |
| MASSP test-retest | 0.513 | 0.528 | 0.467 | 0.461 | 0.532 | 0.508 | 0.488 | 0.509 | 0.391 | 0.419 | 0.533 | 0.536 | 0.431 | 0.464 | 0.428 | 0.402 | 0.433 |
| Freesurfer test-retest | 0.876 | | | | 0.778 | | 0.838 | 0.858 | 0.430 | 0.769 | 0.692 | | | | | | |
| MASSP – Freesurfer | 0.976 | | | | 1.673 | | 1.605 | 1.946 | 5.699 | 1.428 | 1.478 | | | | | | |

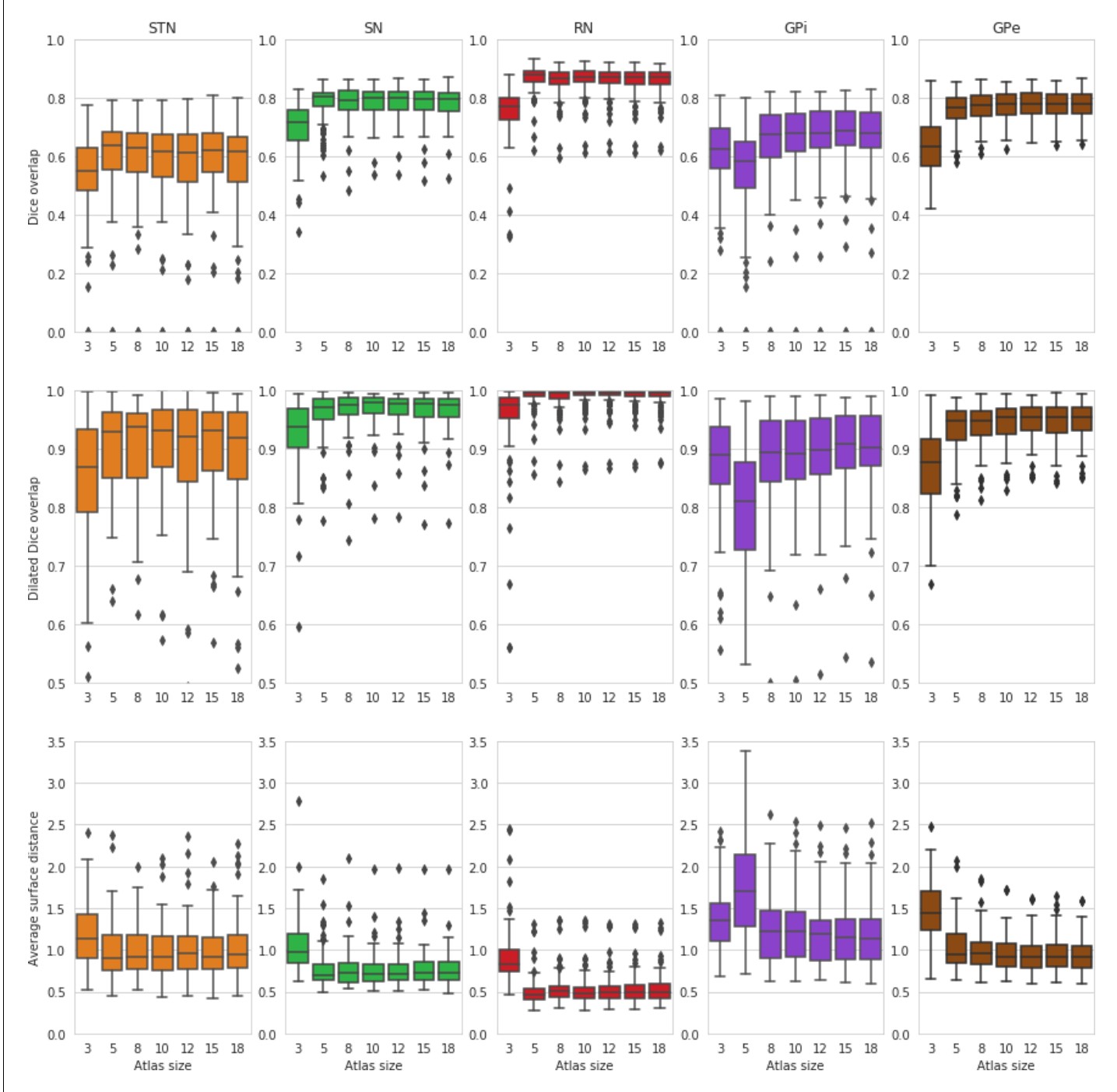

**Figure 6.** MASSP parcellation scores as a function of increasing number of subjects included in the atlas. Scores for the left and right side are computed separately and then combined into box-and-whisker plots.

Welch's t-tests between using 18 subjects or any subset of 8 or more for all structures and measures).

## Biases due to age differences

To more specifically test the influence of age on parcellation accuracy, we defined again six age groups by decade and randomly selected 10 subjects from each group. Each set of subjects was used as priors for the five structures above, and applied to the other age groups. Results are

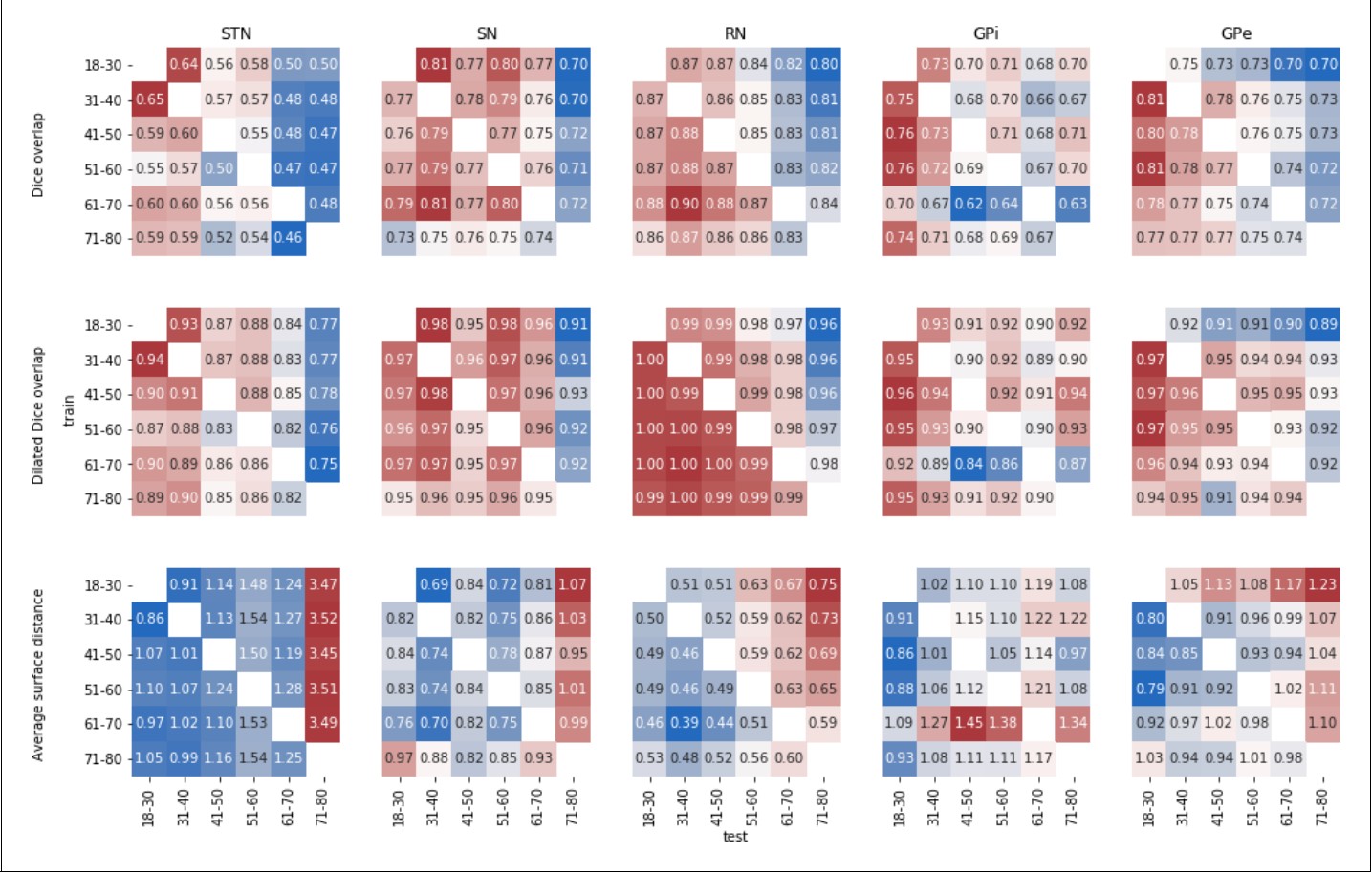

**Figure 7.** MASSP parcellation scores over the lifespan. Each matrix show the average Dice overlap (top), dilated Dice overlap (middle), and average surface distance (bottom) for using one age group as prior ('train') to parcellate another age group ('test').

summarized in *Figure 7*. Examining this age bias, we can see a decrease in performance when parcellating subjects in the range of 60 to 80 years of age. The choice of priors seem to have a limited impact, which varies across structures. In particular, using priors from a similar age group is not always beneficial.

### Bias on individual measures

Finally, we investigated the impact of this decrease in performance in the estimation of anatomical quantities, see *Figure 8*. The bias did affect the morphometric measures of structure volume and thickness, but the effects on the local measure of thickness was reduced compared to the global measure of volume. Quantitative MRI averages were very stable even when age biases are present in the parcellations.

For reference, we report structure volumes, thickness, R1, R2* and QSM values estimated from the entire AHEAD cohort for different age groups, extending our previous work based on manual delineations on a different data set (*Keuken et al., 2017*; *Forstmann et al., 2014*). Results are given in *Table 4*, describing average volumes, thickness, and quantitative MRI parameters for young, middle-aged, and older subjects for the 17 subcortical structures.

### Discussion

Our goal with the MASSP algorithm was to provide a fully automated method to delineate as many subcortical structures as possible on high-resolution structural MRI now available on 7T scanners. We

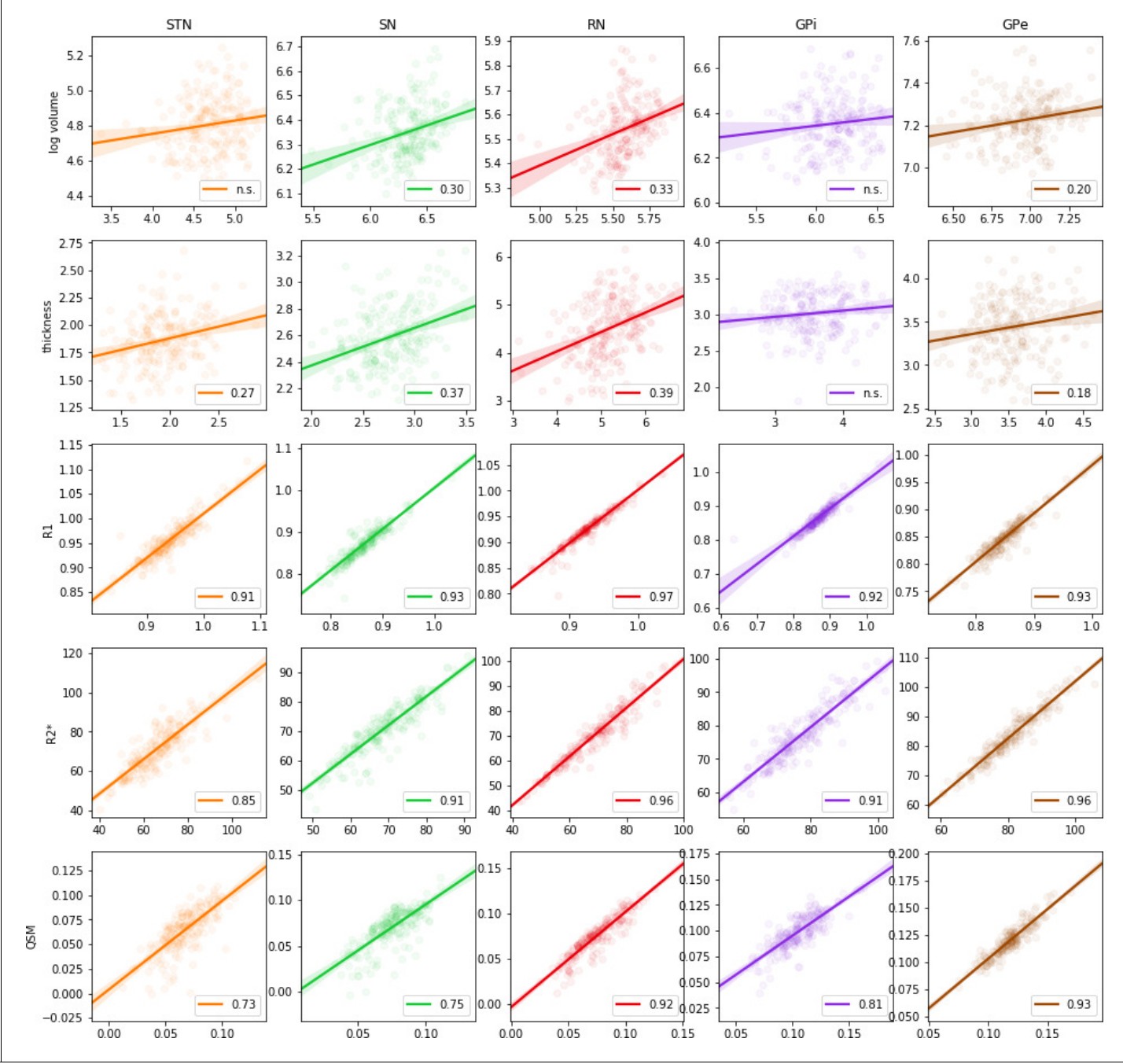

**Figure 8.** Regression of volume (log scale), structure thickness, R1, R2*, and QSM MRI parameters estimated using manual delineations versus MASSP automated parcellations. Circles show individual data points, linear regression is indicated by a straight line, and 95% confidence interval is given as the shaded area. Pearson correlation coefficients are indicated when significant (p-value<0.01).

modeled 17 distinct structures, taking into account location, shape, volume, and quantitative MRI contrasts to provide individual subject parcellations. Based on our results, we can be confident that the automated parcellation technique performs comparably to human experts, providing delineations within one or two voxels of the structure boundaries (dilated Dice overlap over 75% for all structures, including in aging groups). Results were nearly indistinguishable from expert delineations for eight major structures (Str, Tha, 4V, GPe, SN, RN, VTA, ic), and smaller structures retain high levels of overlap, comparable to trained human raters. This parcellation includes the most commonly defined structures (Str, Tha, SN, RN, STN) with overlap scores comparable to those previously

**Table 4.** Mean volume and quantitative MRI values for each age group.

| Age | Str | STN | SN | RN | GPi | GPe | Tha | LV | 3V | 4V | Amg | ic | VTA | fx | PAG | PPN | Cl |
|---|---|---|---|---|---|---|---|---|---|---|---|---|---|---|---|---|---|
| | | | | | | | Volume ($mm^3$) | | | | | | | | | | |
| 18-40 | 10656 | 118 | 566 | 253 | 567 | 1366 | 7112 | 7524 | 1895 | 1391 | 1315 | 4335 | 254 | 1632 | 250 | 193 | 843 |
| 41-60 | 10572 | 124 | 583 | 256 | 586 | 1403 | 7492 | 8850 | 2024 | 1408 | 1363 | 4495 | 264 | 1808 | 255 | 195 | 830 |
| 61-80 | 10734 | 130 | 584 | 260 | 586 | 1397 | 7463 | 9142 | 2023 | 1407 | 1321 | 4407 | 272 | 1910 | 259 | 192 | 829 |
| | | | | | | | Thickness ($mm$) | | | | | | | | | | |
| 18-40 | 5.94 | 1.89 | 2.55 | 4.64 | 3.09 | 3.56 | 8.31 | 4.27 | 2.77 | 4.03 | 4.81 | 4.06 | 1.69 | 2.06 | 1.78 | 1.92 | 1.79 |
| 41-60 | 5.47 | 1.86 | 2.66 | 4.58 | 2.96 | 3.41 | 8.28 | 5.08 | 2.95 | 3.89 | 4.85 | 4.19 | 1.76 | 1.96 | 1.84 | 1.86 | 1.80 |
| 61-80 | 5.22 | 1.83 | 2.60 | 4.11 | 2.92 | 3.22 | 8.28 | 4.90 | 3.18 | 4.06 | 4.73 | 4.19 | 1.80 | 1.97 | 1.90 | 1.95 | 1.82 |
| | | | | | | | qR1 ($Hz$) | | | | | | | | | | |
| 18-40 | 0.647 | 0.949 | 0.857 | 0.928 | 0.868 | 0.850 | 0.761 | 0.332 | 0.346 | 0.274 | 0.546 | 0.906 | 0.819 | 0.714 | 0.654 | 0.779 | 0.650 |
| 41-60 | 0.662 | 0.968 | 0.893 | 0.939 | 0.879 | 0.856 | 0.758 | 0.278 | 0.315 | 0.269 | 0.559 | 0.904 | 0.833 | 0.671 | 0.653 | 0.771 | 0.664 |
| 61-80 | 0.648 | 0.952 | 0.882 | 0.903 | 0.860 | 0.830 | 0.743 | 0.273 | 0.300 | 0.270 | 0.552 | 0.884 | 0.814 | 0.638 | 0.647 | 0.764 | 0.669 |
| | | | | | | | qR2* ($Hz$) | | | | | | | | | | |
| 18-40 | 43.8 | 67.1 | 67.8 | 63.2 | 75.2 | 79.6 | 38.1 | 14.7 | 18.9 | 9.0 | 25.5 | 36.8 | 39.2 | 37.4 | 25.9 | 32.7 | 32.6 |
| 41-60 | 50.4 | 74.1 | 74.1 | 77.1 | 80.2 | 87.9 | 40.3 | 8.4 | 12.4 | 11.7 | 28.1 | 38.7 | 42.8 | 37.4 | 28.0 | 33.4 | 36.9 |
| 61-80 | 51.8 | 77.0 | 72.5 | 73.8 | 77.8 | 87.0 | 40.1 | 8.5 | 10.2 | 12.0 | 30.1 | 39.6 | 52.6 | 35.7 | 28.4 | 34.2 | 35.4 |
| | | | | | | | QSM ($ppm$) | | | | | | | | | | |
| 18-40 | 0.0329 | 0.0609 | 0.0738 | 0.0717 | 0.1015 | 0.1150 | 0.0138 | 0.0130 | 0.0100 | 0.0279 | 0.0036 | −0.0234 | 0.0241 | 0.0079 | 0.0119 | 0.0135 | −0.0122 |
| 41-60 | 0.0400 | 0.0647 | 0.0713 | 0.0829 | 0.0984 | 0.1241 | 0.0134 | 0.0115 | 0.0025 | 0.0234 | 0.0085 | −0.0226 | 0.0201 | 0.0079 | 0.0089 | 0.0099 | −0.0110 |
| 61-80 | 0.0411 | 0.0705 | 0.0610 | 0.0738 | 0.0925 | 0.1249 | 0.0064 | 0.0089 | −0.0034 | 0.0236 | 0.0061 | −0.0243 | 0.0177 | 0.0100 | 0.0039 | 0.0096 | −0.0091 |

reported (*Garzón et al., 2018*; *Visser et al., 2016a*; *Eugenio Iglesias et al., 2013*; *Chakravarty et al., 2013*; *Patenaude et al., 2011*). More importantly, it also includes structures seldom or never before considered in MRI atlases and parcellation methods, such as GPe, GPi, VTA, 3V, 4V, ic, fx, PAG, PPN, Cl. The technique handles structures of varying sizes well, as indicated by dilated overlap and boundary distance. Additional structures can be added, if they can be reliably delineated by expert raters on single-subject MRI at achievable resolutions. Some enhancement techniques such as building a multi-subject template (*Pauli et al., 2018*) or adding a denoising step (*Bazin et al., 2019*) may be beneficial. Co-registration to a high-precision atlas as in *Ewert et al., 2018* may also improve the initial alignment over the MASSP group average template.

Age biases are present both in expert manual delineations and automated parcellation techniques. Age trajectories in volume and quantitative MR parameters indicate systematic shifts in contrast intensities and an increasing variability with age, associated with changing myelination, iron deposition, and brain atrophy (*Draganski et al., 2011*; *Daugherty and Raz, 2013*; *Fjell et al., 2013*; *Keuken et al., 2017*). These changes seem only to impact the parcellation accuracy for age groups beyond 60 years and age-matched priors did not provide specific improvements, thus indicating that an explicit modeling of age effects may be required to further improve parcellation quality in elderly populations. These results also point to exercising caution when applying automated parcellation methods to study morphometry in elderly or diseased populations, where measured differences may include biases. They also point out that while global volume and local thickness are indeed affected by such biases, quantitative MRI measures are much more robust. Note that this bias is likely present is many automated methods, although they have not been systematically investigated due to the extensive manual labor required. Interestingly, biases also exist in expert delineations: when the size or shape of a structure is refined in neuroanatomical studies, experts may become more or less conservative in their delineations. Automated methods provide a more objective measure in such case, as the source of their bias is explicitly encoded in the atlas prior delineations and computational model. Important applications of subcortical parcellation also include deep-brain stimulation surgery (*Ewert et al., 2018*), where the number of structures parcellated by MASSP can

help neurosurgeons orient themselves more easily, although precise targeting will still require manual refinements, especially in neurodegenerative diseases.

We observed that dilated overlap, that is, the overlap of structures up to one voxel, provided a measure of accuracy largely independent of size, for automated or manual delineations. Imprecision in the range of one voxel in the boundary is to be expected due to partial voluming which impacts Dice overlap. The dilated overlap measure is a better representative of performance and indicates that conservative or inclusive versions of the subcortical regions can be obtained by eroding or dilating the estimated boundary by a single voxel. Such masks may be useful when separating functional MRI signals between neighboring nuclei or when locating smaller features inside a structure. Additionally, the Bayesian estimation framework provides voxel-wise probability values, which can also be used to further weight the contribution of each voxel within a region in subsequent analyses.

In summary, our method provides fast and accurate parcellation for subcortical structures of varying size, taking advantage of the high resolution offered by 7T and the specificity of quantitative MRI. The algorithm is based on an explicit model of structures given in a Bayesian framework and is free of tuning parameters. Given a different set of regions of interest or different populations, new priors can be automatically generated and used as the basis for the algorithm. If more MRI contrasts are available, the method can also be augmented to take them into account. The main requirement for the technique is a set of manual delineations of all the structures of interest in a small group of representative subjects. Performance may further improve with the number of included structures, as the number of distinct interfaces increases, refining in particular the intensity priors. In future works, we plan to include more structures or sub-structures and model the effects of age on the priors. We hope that the method, available in open source, will help neuroscience researchers to include more subcortical regions in their structural and functional imaging studies.

## Materials and methods

### Data acquisition

Our parcellation method has been developed for the MP2RAGEME sequence (*Caan et al., 2019*). Briefly, the MP2RAGEME consists of two interleaved MPRAGEs with different inversions and four echoes in the second inversion. Based on these images, one can estimate quantitative MR parameters of R1, R2* and QSM. In this work, we used the following sequence parameters: inversion times TI1,2 = 670 ms, 3675.4 ms; echo times TE1 = 3 ms, TE2,1–4 = 3, 11.5, 19, 28.5 ms; flip angles FA1,2 = 4°, 4°; TRGRE1,2 = 6.2 ms, 31 ms; bandwidth = 404.9 MHz; TRMP2RAGE = 6778 ms; SENSE acceleration factor = 2; FOV = 205×205 x 164 mm; acquired voxel size = 0.70×0.7 x 0.7 mm; acquisition matrix was 292 × 290; reconstructed voxel size = 0.64×0.64 x 0.7 mm; turbo factor (TFE) = 150 resulting in 176 shots; total acquisition time = 19.53 min.

T1-maps were computed using a look-up table (*Marques et al., 2010*). T2*-maps were computed by least-squares fitting of the exponential signal decay over the multi-echo images of the second inversion. R1 and R2* maps were obtained as the inverse of T1 and T2*. For QSM, phase maps were pre-processed using iHARPERELLA (integrated phase unwrapping and background phase removal using the Laplacian) of which the QSM images were computed using LSQR (*Li et al., 2014*). Skull information was removed through creation of a binary mask using FSL's brain extraction tool on the reconstructed uniform T1-weighted image and then applied to the quantitative contrasts (*Smith, 2002*). As all images were acquired as part of a single sequence, no co-registration of the quantitative maps was required (see *Figure 9*).

### Anatomical structure delineations

Manual delineations of subcortical structures were performed by two raters trained by an expert anatomist, according to protocols optimized to use the better contrast or combination of contrasts for each structure and to ensure a consistent approach across raters. The following 17 structures were defined on a group of 10 subjects (average age 24.4, eight female): striatum (Str), thalamus (Tha), lateral, 3rd and 4th ventricles (LV, 3V, 4V), amygdala (Amg), globus pallidus internal segment (GPi) and external segment (GPe), SN, STN, red nucleus (RN), ventral tegmental area (VTA), fornix (fx), internal capsule (ic), periaqueductal gray (PAG), pedunculopontine nucleus (PPN), and claustrum (Cl). Separate masks for left and right hemisphere were delineated except for 3V, 4V, and fx. In the

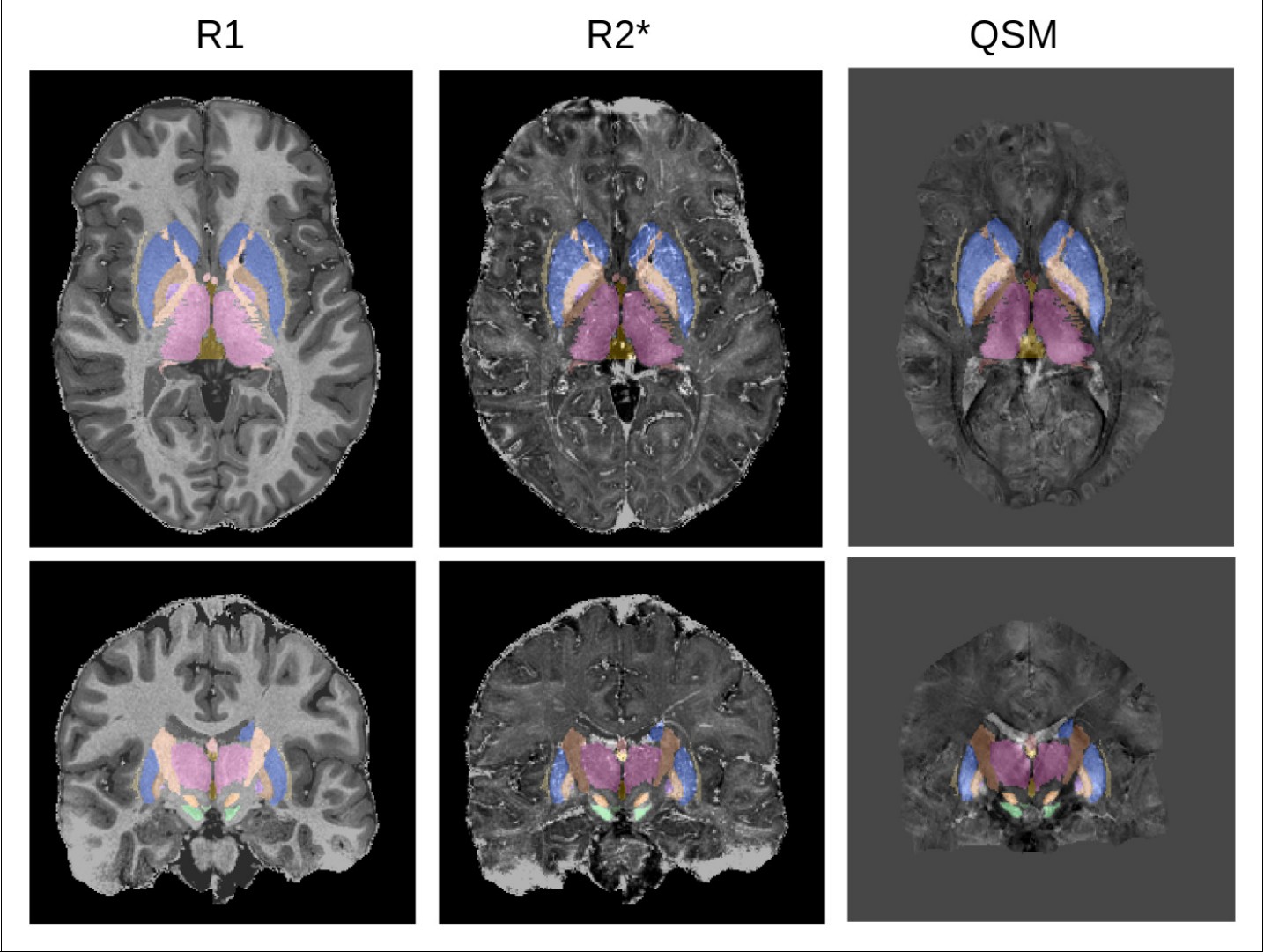

**Figure 9.** MP2RAGEME maps and delineations: quantitative R1 (left), quantitative R2* (middle), QSM (right). Manual delineations for the 17 structures of interest are overlaid on all images.

following the algorithm treats each side separately, resulting in a total of 31 distinct structures (see *Figure 1*).

### Anatomical interface priors

In order to inform the algorithm, we built a series of priors derived from the manual delineations. Each subject was first co-registered to a MP2RAGEME anatomical template built from 105 subjects co-aligned with the MNI2009b atlas (*Fonov et al., 2011*) with the SyN algorithm of ANTs (*Avants et al., 2008*) using successively rigid, affine, and non-linear transformations, high levels of regularization as recommended for the subcortex (*Ewert et al., 2019*) and mutual information as cost function.

The first computed prior is a prior of anatomical interfaces, recording the most likely location of boundaries between the different structures, defined as follows. Given two delineated structures $i, j$, let $\varphi_i, \varphi_j$ be the signed distance functions to their respective boundary, that is, $\varphi_i(x)$ is the Euclidean distance of any given voxel to the boundary of $i$, with a negative sign inside the structure. Then we define the interface $B_{i|j}$ with the distance function $d_{i|j}$:

$$d_{i|j}(x) = \min\left(\varphi_i(x), \varphi_j(x) - \delta, 0\right) \tag{1}$$

where $\delta$ is a scale parameter for the thickness of the interface. These interfaces functions are not symmetrical, as the intensity inside $i$ next to $j$ is generally different from the intensity inside $j$ next to $i$. Based on this definition, the prior for a given interface based on $N$ manual delineations is given by:

$$P(x \in B_{i|j}) \sim \frac{1}{\sqrt{2\pi\sigma_{i|j}^2(x)}}\exp-\frac{1}{2}\frac{\mu_{i|j}^2(x)}{\sigma_{i|j}^2(x)}$$

$$\mu_{i|j}(x) = \frac{1}{N}\sum_{n\in N} d_{i|j,n}(x), \; \sigma_{i|j}(x) = \sqrt{\frac{1}{N}\sum_{n\in N}\left(d_{i|j,n}(x) - \mu_{i|j}(x)\right)^2} + \delta \tag{2}$$

These probability functions are calculated for all possible configurations including $i|i$, which represent the inside of each structure. We thus have a total of $N^2$ functions, but only a few are non-zero at a given voxel $x$, and we may keep only the 16 largest values to account for any number of interfaces in 3D (**Bazin et al., 2007**). Finally, we need to scale the prior to be globally consistent with the priors below by assuming that the 95th percentile of the highest kept $P(x \in B_{i|j})$ values have a probability of 0.95. The scale parameter $\delta$ is set to one voxel, representing the expected amount of partial voluming. The resulting interface prior is shown in **Figure 10A**.

### Anatomical skeleton priors

Next, we defined priors for the skeleton of each structure, representing their essential shape regardless of exact boundaries (**Blum, 1973**). As we are mostly interested in the most likely components of the skeleton or medial axis $S_i$, we follow a simple method to estimate its location:

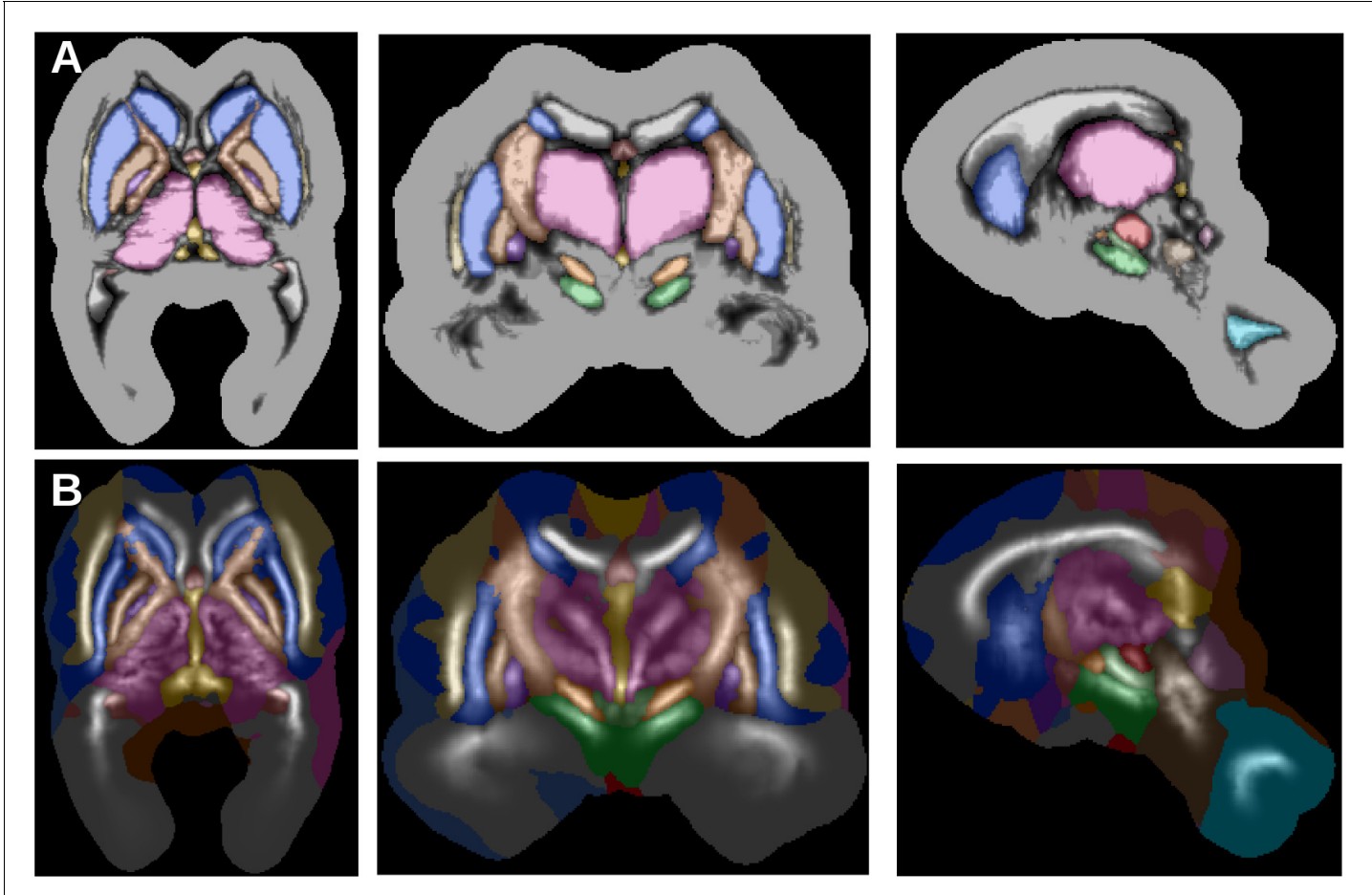

**Figure 10.** Anatomical interface (**A**) and skeleton (**B**) priors derived from the 10 manually delineated subjects.

$$S_i = \left\{ x, |\nabla \varphi_i(x)| < \frac{1}{2} \right\} \tag{3}$$

We define as $s_i(x)$ the signed distance function of this discrete skeleton, and define prior probabilities as above:

$$P(x \in S_i) \sim \frac{1}{\sqrt{2\pi\sigma_i^2(x)}} \exp -\frac{1}{2}\frac{\mu_i^2(x)}{\sigma_i^2(x)}$$

$$\mu_i(x) = \frac{1}{N}\sum_{n\in N} s_{i,n}(x) \; \sigma_i(x) = \sqrt{\frac{1}{N}\sum_{n\in N}\left(s_{i,n}(x) - \mu_i(x)\right)^2} + \delta \tag{4}$$

The skeletons are defined inside each structure, which implies $P(x \in S_i) \leq P(x \in B_{i|i})$. To respect this relationship, we scale $P(x \in S_i)$ with the same factor as $P(x \in B_{i|i})$ but use $\sqrt{P(x \in S_i)}$ when combining probabilities during the estimation stage. The obtained anatomical skeleton priors are given on *Figure 10B*.

## Interface intensity priors

While anatomical priors already provide rich information, they are largely independent of the underlying MRI. From the co-aligned quantitative MRI maps and manual delineations, we defined intensity priors for every interface $i|j$, in the form of intensity histograms to ensure a flexible representation of intensity distributions. Given a quantitative contrast $R_n(x)$, we built a histogram $H_{i|j,n}$ for each subject $n$ and interface $i|j$. Histograms have 200 bins covering the entire intensity range within a radius of 10 mm from any of the delineated structures. To obtain an average histogram, we combine each histogram with a weighting function $w_n(x)$ giving the likelihood of the subject's intensity measurement compared to the group:

$$w_{i|j,n}(x) = P(x \in B_{i|j})\frac{1}{\sqrt{2\pi\sigma_R^2}}\exp -\frac{1}{2}\frac{(R_n(x) - \mu_R(x))^2}{\sigma_R(x)^2} \tag{5}$$

where $\mu_R(x)$ is the median of the $R_n(x)$ values at $x$, and $\sigma_R(x)$ is 1.349 times the inter-quartile range of $R_n(x)$. These are robust estimators of the mean and standard deviation, used here to avoid biases by intensity outliers. To further combine the R1, R2*, and QSM contrasts we take the geometric mean of the histogram probabilities: $H_{i|j}(x) = \prod_R H_{i|j}(R(x))^{1/3}$.

## Global volume priors

The last type of priors extracted from manual delineations are volume priors for each of the structure. Here, we assume a log-normal distribution for the volumes $V_i$ and simply estimate the mean $\mu_{V,i}$ and standard deviation $\sigma_{V,i}$ of $\log V_{i,n}$ over the subjects.

## Voxel-wise posterior probabilities

When parcellating a new subject, we first co-register its R1, R2*, and QSM maps jointly to the template and use the inverse transformation to deform the anatomical priors into subject space. Then we derive voxel-wise posteriors as follows:

$$P(x \in B_{i|j}|R(x)) \sim P(x \in B_{i|j})H_{i|j}(x) \text{ if } i \neq j$$
$$\text{and}$$
$$P(x \in B_{i|i}|S_i(x), R(x)) \sim \max(P(x \in B_{i|i}), P(x \in S_i)^{1/2})H_{i|i}(x) \tag{6}$$

Once again we should compute all possible combinations, but due to the multiplication of the priors we can restrict ourselves to the 16 highest probabilities previously estimated. To balance the contribution of the anatomical priors and the intensity histograms, we also need to normalize the intensity priors sampled on the subject's intensities. We use the same approach, namely assuming that the 95th percentile of the highest kept $H_{i|j}(x)$ values have a probability of 0.95, separately for each contrast. The voxel-wise parcellation and posteriors obtained are shown in *Figure 11A*.

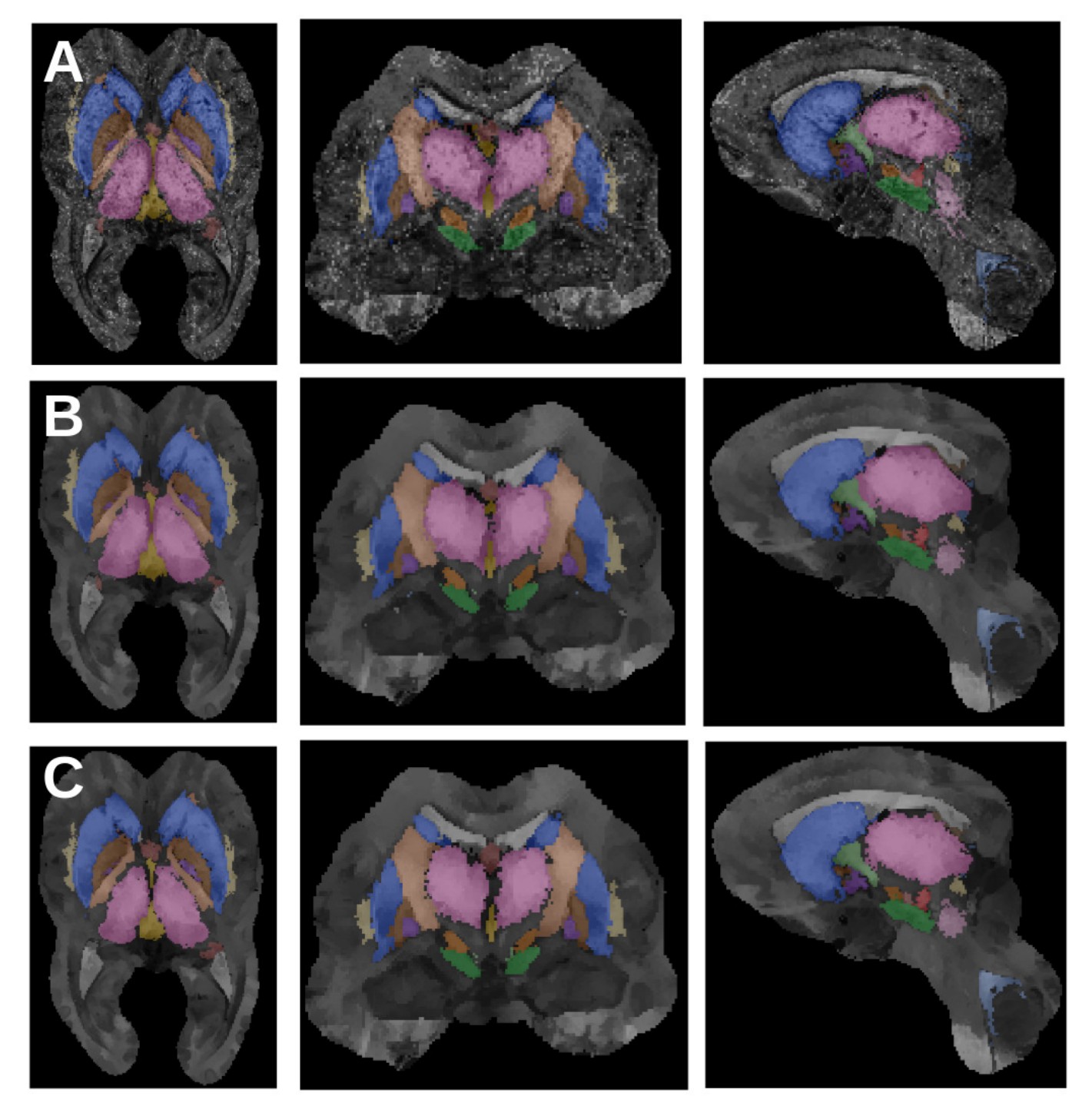

**Figure 11.** Successive parcellation results: (A) voxel-wise posteriors and parcellation, (B) diffused posteriors and parcellation, (C) topology-corrected posteriors and final region-growing parcellation.

### Markovian diffusion

The voxel-wise posteriors are independent from each other and do not reflect the continuous nature of the structures. The next step is to combine information from neighboring voxels. We define a sparse Markov Random Field model for the posteriors:

$$P(x \in B_{i|j}|R,S,C) = \sum_{y \in C(x)} P(y \sim x|R)P(y \in B_{i|j}|R,S,C) \tag{7}$$

with $P(y \sim x|R) = \prod_R \exp -(R(y) - R(x))^2/2\sigma_R^2$, where $\sigma_R$ is the median of the standard deviations $\sigma_{i|j,R}$ of the contrast histograms $H_{i|j}(R(x))$. The neighborhood $C(x)$ is defined as $x$ itself and the four 26-connected neighboring voxels with highest probability $P(y \sim x|R)$, thus representing the neighbors most likely to be connected to $x$. The model is similar to a diffusion process and can be estimated with an iterated conditional modes (ICM) approach, updating sequentially the probabilities (**Bazin and Pham, 2007**):

$$P(x \in B_{i|j}|R,S,C) \leftarrow \sum_{y \in C(x)} P(y \sim x|R)P(y \in B_{i|j}|R,S,C) \tag{8}$$

from the initial voxel-wise posteriors until the ratio of changed parcellation labels decreases below 0.1, typically within 50–80 iterations. The diffused probabilities and parcellation are shown in **Figure 11B**.

## Topology correction

The final step of the parcellation algorithm takes a global view of the individual structures, growing from the highest posterior values inside toward the boundaries. This region growing approach makes the implicit assumption that posterior maps should be monotonically decreasing from inside to outside, which is not necessarily the case. Therefore, we perform first a topology correction step on the individual structure posteriors $P(x \in i|R,B,S,C) = \max_{i|j} P(x \in B_{i|j}|R,S,C)$ with a fast marching algorithm (**Bazin and Pham, 2007**). While the corrected posterior is very similar to the original one (see **Figure 11C**), it ensures that all regions obtained by growing to a threshold have spherical object topology.

## Anatomical region growing

Last, we turn the posteriors into optimized parcellations, by growing them concurrently (to avoid overlaps) until the target volume for each structure is reached. Given the volume $V_i(R,B,S,C)$ of the parcellation of the diffused and topology-corrected posteriors, we define the following target volume:

$$\hat{V}_i = P(V_i|\mu_{V,i}, \sigma_{V,i})V_i + (1 - P(V_i|\mu_{V,i}, \sigma_{V,i}))\exp \mu_{V,i} \tag{9}$$

taking a weighted average of the volume estimated from the data and the prior volume. This approach ensures that even in extreme cases where some structures have low posteriors, they are still able to grow to a plausible size. The region growing algorithm is driven from the most likely voxels, defined as $P(x \in i|R,B,S,C) - \max_{j \neq i} P(x \in j|R,B,S,C)$, and further modulated to follow isocontours of the skeleton prior:

$$P(x \leftarrow y) \sim \quad P(y \in i|R,B,S,C) - \max_{j \neq i} P(y \in j|R,B,S,C) \\ -|P(y \in S_i) - P(x \in S_i)| \tag{10}$$

Directionality of internal structures is a useful tool for understanding mechanical function in bones (**Maquer et al., 2015**). Here, we adapt this concept by using the skeleton isocontours as a representation of internal directionality, maintaining the intrinsic shape of structures. Thus, voxels with highest probability compared to the other structures and with similar distance to the internal skeleton are preferentially selected. The final parcellation is given in **Figure 11C**.

## Validation metrics

To validate the method against manual expert delineations, we compared the MASSP results and the expert delineations with the following three measures:

1. The Dice overlap coefficient (**Dice, 1945**) $D(A,B) = \frac{2A \cap B}{A+B}$, which measures the strict overlap between voxels in both delineation;

2. The dilated overlap coefficient $dD(A,B) = \frac{A \cup d(B) + B \cup d(A)}{A+B}$, where $d(.)$ is a dilation of the delineation by one voxel, which measures the overlap between delineations allowing for one voxel of uncertainty;

3. The average surface distance $asd(A,B)$, measuring the average distance between voxels on the surface boundary of the first delineation to the other one and reciprocally, which measures the distance between both delineations.

We computed all three measures for the manual delineations from the two independent raters, as well as the ratio of overlaps (automated over manual) and distances (manual over automated) to compare both performances, as detailed in the Results section.

## Comparisons with other automated methods

To assess the performance of MASSP compared to existing parcellation tools, we ran Freesurfer (*Fischl et al., 2002*), FSL FIRST (*Patenaude et al., 2011*) and a multi-atlas registration approach (co-registering 9 of the 10 manually delineated subjects on the remaining one with ANTs [*Avants et al., 2008*] and labeling each structure by majority voting, similarly to the MAGeT Brain approach of *Chakravarty et al., 2013*). Freesurfer and FIRST were run on the skull-stripped R1 map, while the multi-atlas approach used all three R1, R2*, and QSM contrasts. All methods were compared in terms of Dice overlap, dilated overlap and average surface distance. We also assessed the presence of a systematic volume bias, defined as the average of the signed difference of the estimated structure volume to the manually delineated volume, normalized by the manually delineated volume.

## Application to new MRI contrasts

Before applying MASSP to unseen contrasts, we need to convert its intensity prior histograms $H_{i|j,R}$ to the new intensities. In order to perform this mapping, we first created a groupwise median of the HCP subjects, by co-registering every subject to the MASSP template using ANTs with non-linear registration and both T1w, T2w contrasts matched to the template's R1 and R2* maps. The histogram bins are then updated as follows:

$$H_{\text{bin},i|j,R} \equiv \sum_{x|R(x) \in \text{bin}} P(x \in B_{i|j}) H_{i|j,R1} H_{i|j,R2*} H_{i|j,QSM} \tag{11}$$

adding the joint probability of the quantitative contrasts weighted by their importance for each interface to define the new intensity histograms. This model is essentially projecting the joint likelihood of the MASSP contrasts onto the new contrasts, assuming that the co-registration between the two is accurate enough. With these new histograms, we compared the test-retest reliability and overall agreement of MASSP with Freesurfer parcellations included in the HCP pre-processed data set.

## Measurement of structure thickness

Finally, when comparing derived measures obtained over the lifespan with MASSP compared to manual delineations, we explored the utility of a shape thickness metric, based on the medial representation. Given the signed distance function $\varphi_i$ of the structure boundary and $s_i$ of the structure skeleton, the thickness is given by:

$$th_i(x) = 2(s_i(x) - \varphi_i(x)) \tag{12}$$

Like in cortical morphometry, thickness is a local measure, defined everywhere inside the structure, and expected to provide additional information about anatomical variations. Indeed, a similar measure of shape thickness has recently been able to highlight subtle anatomical changes in depression (*Ho et al., 2020*).

## Software implementation

The proposed method, Multi-contrast Anatomical Subcortical Structure Parcellation (MASSP), has been implemented as part of the Nighres toolbox (*Huntenburg et al., 2018*), using Python and Java for optimized processing. The software is available in open source from (release-1.3.0) and . A

complete parcellation pipeline is included with the Nighres examples. Computations take under 30 min per subject on a modern workstation.

## Acknowledgements

We thank Josephine Groot, Nikita Berendonk, Nicky Lute for their help collecting the AHEAD database, and Wietske van der Zwaag and Matthan Caan for their help in setting up the MP2RAGEME sequence. We also thank Steven Miletić and Dagmar Timmann for stimulating discussions around this topic, and all undergraduate students who contributed to the manual delineations. This work was supported by a NWO Vici grant (BF) and a NWO STW grant (AA, BF). HCP data were provided by the Human Connectome Project, WU-Minn Consortium (Principal Investigators: David Van Essen and Kamil Ugurbil; 1U54MH091657) funded by the 16 NIH Institutes and Centers that support the NIH Blueprint for Neuroscience Research; and by the McDonnell Center for Systems Neuroscience at Washington University.

## Additional information

### Funding

| Funder | Grant reference number | Author |
| --- | --- | --- |
| Nederlandse Organisatie voor Wetenschappelijk Onderzoek | VICI | Birte U Forstmann |
| Nederlandse Organisatie voor Wetenschappelijk Onderzoek | STW | Anneke Alkemade Birte U Forstmann |

The funders had no role in study design, data collection and interpretation, or the decision to submit the work for publication.

### Author contributions

Pierre-Louis Bazin, Conceptualization, Software, Formal analysis, Validation, Investigation, Visualization, Methodology, Writing - original draft; Anneke Alkemade, Data curation, Formal analysis, Validation, Investigation, Methodology, Writing - review and editing; Martijn J Mulder, Resources, Data curation, Software, Methodology, Writing - review and editing; Amanda G Henry, Conceptualization, Methodology, Writing - review and editing; Birte U Forstmann, Conceptualization, Resources, Formal analysis, Supervision, Funding acquisition, Project administration, Writing - review and editing

### Author ORCIDs

Pierre-Louis Bazin https://orcid.org/0000-0002-0141-5510
Anneke Alkemade http://orcid.org/0000-0002-3234-353X
Amanda G Henry http://orcid.org/0000-0002-2923-4199
Birte U Forstmann http://orcid.org/0000-0002-1005-1675

### Ethics

Human subjects: Informed consent and consent to publish, including consent to publish anonymized imaging data, was obtained for all subjects. Ethical approval was obtained with the University of Amsterdam Faculty of Social and Behavioral Sciences LAB Ethics Review Board, with ERB number 2016-DP-6897.

### Decision letter and Author response

Decision letter https://doi.org/10.7554/eLife.59430.sa1
Author response https://doi.org/10.7554/eLife.59430.sa2

# Additional files

## Supplementary files
• Transparent reporting form

## Data availability

The tool presented in this article is available in open source on Github (https://github.com/nighres/nighres). The atlases necessary to run the algorithm have been deposited on the University of Amsterdam FigShare (https://doi.org/10.21942/uva.12074175.v1 and https://doi.org/10.21942/uva.12301106.v2). A single sample subject data set has been deposited on the University of Amsterdam FigShare (https://doi.org/10.21942/uva.12280316.v2). All the measurements used to generate the figures included in the article have been deposited on the University of Amsterdam FigShare (https://doi.org/10.21942/uva.12452444.v1).

The following previously published datasets were used:

| Author(s) | Year | Dataset title | Dataset URL | Database and Identifier |
|---|---|---|---|---|
| Alkemade A, Mulder MJ, Groot JM, Isaacs BR, van Berendonk N, Lute N, Isherwood SJ, Bazin P-L, Forstmann BU | 2020 | The Amsterdam Ultra-high field adult lifespan database (AHEAD): A freely available multimodal 7 Tesla submillimeter magnetic resonance imaging database | https://doi.org/10.21942/uva.10007840.v1 | FigShare / University of Amsterdam / Amsterdam University of Applied Sciences, 10.21942/uva.10007840.v1 |
| Van Essen DC, Smith SM, Barch DM, Behrens TEJ, Yacoub E, Ugurbil K | 2013 | The WU-Minn Human Connectome Project | https://www.humanconnectome.org/study/hcp-young-adult | WU-Minn HCP Retest Data, hcp-young-adult |

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
