## [Decision Letter]

**Acceptance summary:**

There was unanimous agreement that this method has strong potential to be a new "workhorse" tool in human neuroimaging that could substantially advance our ability to measure brain structures that are largely overlooked due to problems with segmentation.

**Decision letter after peer review:**

Thank you for submitting your article "Multi-contrast Anatomical Subcortical Structures Parcellation" for consideration by *eLife*. Your article has been reviewed by three peer reviewers, including Timothy Verstynen as the Reviewing Editor and Reviewer #1, and the evaluation has been overseen by Michael Frank as the Senior Editor. The following individual involved in review of your submission has agreed to reveal their identity: Wolf-Julian Neumann (Reviewer #2).

The reviewers have discussed the reviews with one another and the Reviewing Editor has drafted this decision to help you prepare a revised submission.

Summary:

In this study, Bazin and colleagues propose a novel segmentation algorithm for parcelling subcortical regions of the human brain that was developed from multiple MRI measures derived from the M2RAGEME sequence acquired on a 7T MRI system. The key advancement of this approach is a reliable segmentation of more subcortical areas (17 regions) in native space than what is possible with currently available methods. The authors validate their algorithm by comparing against age-related measures.

This manuscript was reviewed by three experts in the field, who found that this method has strong potential to be a new "workhorse" tool in human neuroimaging that could substantially advance our ability to measure brain structures that are largely overlooked due to problems with segmentation. The main criticisms of the work are largely centered on how the method is evaluated and implemented, rather than fundamental concerns with the validity of the method itself.

Essential revisions:

1) Benchmarks.

All three reviewers had concerns about the nature of the tests for the new method.

Reviewer 1 was particularly concerned that, while a critical advancement of this method is the ability to segment many more regions than previous subcortical atlases, there are still many regions that overlap with existing segmentation tools. Knowing how the reliability of this new approach compares to previous automatic segmentation methods is crucial in being able to know how to trust the overall reliability of the method. The authors should make a direct benchmark against previous methods where they have overlap.

Reviewer 2 thought that the work would certainly benefit from an additional step of out-of-center / out-of-cohort validation analysis. Though they had no serious concern that performance would be unsatisfactory, it would still highlight the extensibility of the method.

Reviewer 3 shared a similar concern, pointing out that automatized methods are usually sensitive to the number of subjects used to build the parcellation, with results from a bigger training cohort being potentially more robust and generalizable. One of the strongest points of the automated method presented in this paper is the adoption of a Bayesian approach, which usually works efficiently for small sample sizes and allows to update previous results when new data comes. Still, it could be highly illustrative to show the performance of the current method depending on the initial training size. From the same set of delineations of the 105 subjects used to test the age bias, what if the authors show the predicted performance from generating the priors on a training set varying its size?

2) Aging analysis.

All three reviewers were confused as to the purpose for and implementation of the aging analysis included in the paper.

Reviewer 1 said that the analysis of the aging effects on the segmentations seemed oddly out of place. It wasn't clear if this is being used to vet the effectiveness of the algorithm (i.e., its ability to pick up on patterns of age-related changes) or the limitations of the algorithm (i.e., the segmentation effectiveness decreases in populations with lower across-voxel contrast). What exactly is the goal with this analysis? Also, why is it limited to only a subset of the regions output from the algorithm?

Reviewer 2 thought that the most important limitation, as acknowledged by the authors, is the bias from anatomical variation through age or disease. The algorithm is shown to be affected by age and most certainly will be affected by contrast and size changes in neurodegenerative disorders. Broader benchmark tests, as proposed above, would likely address this concern.

Reviewer 3 pointed out that, from Figure 4, it is clear how estimated Dice coefficients decrease with age. As it is well noted by the authors, this is likely caused due to the fact that the priors were built from 10 subjects that had an average age of 24.4 years and thus, the highest predicted performance rates are reflected for subjects whose age range (18-40) lies around this average prior age. The authors mentioned in the paper that they plan on modelling the effects of age in the priors in future work. However, they could already address this question in the current work. Since the data used to test this age bias has already been manually delineated, what if the authors generate new priors for this set of delineations, including subjects from all ages, and test whether the predicted Dice coefficients still depend on age, in the same way as was done in Figure 4?

3) Clarity of the algorithm.

Reviewers 1 and 3 had concerns about details of the algorithm itself.

Reviewer 1 thought that, because of the difficulty of the parcellation problem, the algorithm being used is quite complex. The authors do a good job showing the output of each stage of the process (Figure 7 and Figure 8), but it would substantially help general readers to have a schematic of the logic of the algorithm itself.

Reviewer 3 pointed out that it is unclear what is the value for the scale parameter δ that appears in the priors? Is that a free parameter? If so, do results change when this parameter varies? This seems to be a critical aspect of the process (at least insofar as the precision of the results).

[Editors' note: further revisions were suggested prior to acceptance, as described below.]

Thank you for resubmitting your article "Multi-contrast Anatomical Subcortical Structures Parcellation" for consideration by *eLife*. Your revised article has been reviewed by three peer reviewers, including Timothy Verstynen as the Reviewing Editor and Reviewer #1, and the evaluation has been overseen by Michael Frank as the Senior Editor. The following individual involved in review of your submission has agreed to reveal their identity: Wolf-Julian Neumann (Reviewer #2).

The reviewers have discussed the reviews with one another and the Reviewing Editor has drafted this decision to help you prepare a revised submission.

Summary:

The reviewers felt that the revised manuscript is much stronger and focused. There remains only a minor point raised by reviewer #2 that we ask you address.

Essential revisions:

Reviewer #2 (see below) would like to see a more elaborate comparison of the STN segmentation to previous recent methods, given that these sorts of subcortical parcellation methods are likely to be of interest to DBS researchers. While a comparison to the Ewert method (or similar) would be nice, a brief discussion of a subjective comparison of these results would be sufficient to address this concern.

Reviewer #1:

The authors have adequately addressed all of my concerns. Well done.

Reviewer #2:

This is a revised manuscript on a 7T based segmentation approach. My main concern in the primary submission was that at this point the complexity of the data acquisition in combination with the computational approach makes it relatively niche. Other concerns regarded additional validation runs and aging, which the authors have now provided. My feeling is that the authors have made an effort to address the major points raised in the first revision.

There is one minor point that still remains puzzling for me. The fact that the STN is delineated so poorly, when compared to other structures. Given its use as a target in hundreds of thousands of patients for deep brain stimulation, automatized STN segmentation has been validated. The authors cited Ewert et al., which combines a simple normalization with an atlas in MNI space and reached similar and better performance when compared to the presented results here. I would have liked to see the results from this paper reproduced as a comparison here, but if the authors decide not to, I would at least like to invite the authors to discuss why the STN is troublesome for the algorithm and how this could affect use for surgical planning in the future, when 7T becomes available in clinics.

Additionally, I found this a little strange: Figure 3 caption: compared to the most expert of the two human raters.

Reviewer #3:

The authors have addressed all the concerns that I had in the previous version of the manuscript. Important changes in this revision included a benchmark against existing automated parcellation tools, a validation analysis using a test-retest sample from the Human Connectome Project and a thorough examination of training sample size and age biases. All of these changes have significantly increased the quality of the paper and more importantly, provided more clarity and evidence for the benefits of the proposed algorithm for subcortical parcellation. As a consequence, I am more than pleased to recommend the current version of the manuscript for publication.

---

## [Author Response]

Essential revisions:1) Benchmarks.All three reviewers had concerns about the nature of the tests for the new method.Reviewer 1 was particularly concerned that, while a critical advancement of this method is the ability to segment many more regions than previous subcortical atlases, there are still many regions that overlap with existing segmentation tools. Knowing how the reliability of this new approach compares to previous automatic segmentation methods is crucial in being able to know how to trust the overall reliability of the method. The authors should make a direct benchmark against previous methods where they have overlap.

We agree with reviewer 1 that comparing to other previously published methods is useful, as few of them provide tables of validation scores. We added a comparison with three popular tools: a multi-atlas registration scheme building directly on our manual delineations, FSL First, and Freesurfer. The results are summarized in a new Table 2, and detailed further in the Materials and methods and Results sections.

In Results:

“Comparison to other automated methods

To provide a basis for comparison, we applied other freely available methods for subcortical structure parcellation to the same ten subjects. MASSP performs similarly or better than Freesurfer, FSL First and a multi-atlas registration using ANTs. Multi-atlas registration provides high accuracy in most structures as well, but is biased toward under-estimating the size of smaller and elongated structures where overlap is systematically reduced across the individual atlas subjects. Multi-atlas registration is also quite computationally intensive when using multiple contrasts at high resolution. Finally, MASSP provides many more structures than Freesurfer and FSL First, and can be easily applied to new structures based on additional manual delineations.”

In Materials and methods:

“Comparisons with other automated methods

To assess the performance of MASSP compared to existing parcellation tools, we ran Freesurfer (Fischl et al., 2002), FSL First (Patenaude et al., 2011) and a multi-atlas registration approach (coregistering 9 of the 10 manually delineated subjects on the remaining one with ANTs (Avants et al., 2008) and labelling each structure by majority voting, similarly to the MAGeT Brain approach of Chakravarty et. al (2013)). Freesurfer and First were run on the skull-stripped R1 map, while the multi-atlas approach used all three R1, R2* and QSM contrasts. All methods were compared in terms of Dice overlap, and we also assessed the presence of a systematic volume bias, defined as the average of the signed difference of the estimated structure volume to the manually delineated volume, normalized by the manually delineated volume.”

Reviewer 2 thought that the work would certainly benefit from an additional step of out-of-center / out-of-cohort validation analysis. Though they had no serious concern that performance would be unsatisfactory, it would still highlight the extensibility of the method.

We thank reviewer 2 for this thought, which led us to define an appropriate methodology to expand the algorithm to other contrasts. While MASSP is based on quantitative MRI, it is not entirely straightforward to simulate the contrast of various MR sequences based on these, as other sources of contrasts and artifacts often influence the imaging. Instead, we took a data-driven approach, coregistering templates with the contrasts of interest and building a statistical mapping of intensities for each defined interface in the MASSP atlas. With this approach, we were able to successfully parcellate data from the Human Connectome Project. We made the following additions to the Materials and methods and Results sections:

In Results:

“Application to new MRI contrasts

Quantitative MRI has only become recently applicable in larger studies, thanks in part to the development of integrated multi-parameter sequences (Weiskopf et al., 2013; Caan et al., 2019). Many data sets, including large-scale open databases, use more common T1- and T2-weighted MRI. In order to test the applicability of MASSP to such contrasts, we obtained the test-retest subset of the Human Connectome Project (HCP, Van Essen et al., 2013) and applied MASSP to the 45 preprocessed and skull-stripped T1- and T2-weighted images from each of the two test and retest sessions. While performing manual delineations on the new contrasts would be preferable, the model is already rich enough to provide stable parcellations. Test-retest reproducibility is similarly high for MASSP and Freesurfer, and are generally in agreement, see Figure 5 and Table 3.”

In Materials and methods:

“Application to new MRI contrasts

Before applying MASSP to unseen contrasts, we need to convert its intensity prior histograms Hi|j,R to the new intensities. In order to perform this mapping, we first created a groupwise median of the HCP subjects, by co-registering every subject to the MASSP template using ANTs with nonlinear registration and both T1w, T2w contrasts matched to the template's R1 and R2* maps. […] With these new histograms, we compared the test-retest reliability and overall agreement of MASSP with Freesurfer parcellations included in the HCP preprocessed data set.”

Reviewer 3 shared a similar concern, pointing out that automatized methods are usually sensitive to the number of subjects used to build the parcellation, with results from a bigger training cohort being potentially more robust and generalizable. One of the strongest points of the automated method presented in this paper is the adoption of a Bayesian approach, which usually works efficiently for small sample sizes and allows to update previous results when new data comes. Still, it could be highly illustrative to show the performance of the current method depending on the initial training size. From the same set of delineations of the 105 subjects used to test the age bias, what if the authors show the predicted performance from generating the priors on a training set varying its size?

We thank Reviewer 3 for this suggestion. We used the delineations of the five subcortical structures defined on the entire cohort to define atlases of increasing size, from 3 to 18 subjects. Prior work from us and others (e.g. Bazin et al., 2008; Iglesias et al., 2013) have shown that Bayesian approaches are generally very efficient with regard to the size of the training cohort, and we found indeed that performance stabilized for all structures at 8 subjects. We added the experiment to the manuscript as follows in Results:

“Biases due to atlas size

[…] First, we investigated the impact of atlas size. We randomly assigned half of the subjects from each decade to two groups, and built atlas priors from subsets of 3, 5, 8, 10, 12, 15, and 18 subjects from the first group. The subjects used in the atlas were taken randomly from each decade (18-30, 31-40, 41-50, 51-60, 61-70, 71-80), so as to maximize the age range represented in each atlas. Atlases of increasing size were constructed by adding subjects to previous atlases, so that atlases of increasing complexity include all subjects from simpler atlases. Results applying these atlases to parcellate the second group are given in Figure 6. As in previous studies (Iglesias et al., 2013; Bazin et al., 2008), performance quickly stabilized with atlases of more than five subjects (no significant difference in Welch's t-tests between using 18 subjects or any subset of 8 or more for all structures and measures).”

2) Aging analysis.All three reviewers were confused as to the purpose for and implementation of the aging analysis included in the paper.Reviewer 1 said that the analysis of the aging effects on the segmentations seemed oddly out of place. It wasn't clear if this is being used to vet the effectiveness of the algorithm (i.e., its ability to pick up on patterns of age-related changes) or the limitations of the algorithm (i.e., the segmentation effectiveness decreases in populations with lower across-voxel contrast). What exactly is the goal with this analysis? Also, why is it limited to only a subset of the regions output from the algorithm?Reviewer 2 thought that the most important limitation, as acknowledged by the authors, is the bias from anatomical variation through age or disease. The algorithm is shown to be affected by age and most certainly will be affected by contrast and size changes in neurodegenerative disorders. Broader benchmark tests, as proposed above, would likely address this concern.Reviewer 3 pointed out that, from Figure 4, it is clear how estimated Dice coefficients decrease with age. As it is well noted by the authors, this is likely caused due to the fact that the priors were built from 10 subjects that had an average age of 24.4 years and thus, the highest predicted performance rates are reflected for subjects whose age range (18-40) lies around this average prior age. The authors mentioned in the paper that they plan on modelling the effects of age in the priors in future work. However, they could already address this question in the current work. Since the data used to test this age bias has already been manually delineated, what if the authors generate new priors for this set of delineations, including subjects from all ages, and test whether the predicted Dice coefficients still depend on age, in the same way as was done in Figure 4?

We thank the reviewers for their detailed comments and suggestions. Indeed, we realize that the aging analysis was not clearly motivated, and that the experiment of Figure 4 was not very informative. Our goal here was to assess the ability of the algorithm to parcellate data from older subjects, a step seldom taken when validating algorithms. In this revision we replaced the experiment of Figure 4 by a more thorough test of biases: using only the five structures for which we have a complete set of delineations, we tested systematically the impact of atlas age range versus testing age group. The results, reproduced below, indicate that while there is a decrease in performance in subjects above 60 year old, the decrease appears largely independent of the choice of the atlas age group. Thus we conclude that the method accuracy does decrease in subjects above 60 year old and likely in patients with neurodegenerative disease. This effect is not merely a reflection of bias in the atlas, but rather the expression of increasing variability in brain MRI. Note however that the following experiment on estimating morphometry and quantitative MRI from automated versus manual parcellations shows a good stability of local measures across age groups, even if structure boundaries and volumes are not perfectly estimated.

In Results:

“Biases due to age differences

To more specifically test the influence of age on parcellation accuracy, we defined six age groups by decade and randomly selected 10 subjects from each group. Each set of subjects was used as priors for the five structures above, and applied to the other age groups. Results are summarized in Figure 7. Examining this age bias, we can see a decrease in performance when parcellating subjects in the range of 60 to 80 years of age. The choice of priors seems to have a limited impact, which varies across structures. Interestingly, using priors from a similar age group is not particularly beneficial.”

3) Clarity of the algorithm.Reviewers 1 and 3 had concerns about details of the algorithm itself.Reviewer 1 thought that, because of the difficulty of the parcellation problem, the algorithm being used is quite complex. The authors do a good job showing the output of each stage of the process (Figure 7 and Figure 8), but it would substantially help general readers to have a schematic of the logic of the algorithm itself.

We thank reviewer 1 for this valuable suggestion. We added the following explanatory diagram at the beginning of Results:

MASSP uses a data set of ten expert delineations as a basis for its modeling. From the delineations, an atlas of interfaces between structures, shape skeletons, and interface intensity histograms are generated, and used as prior in a multiple-step non-iterative Bayesian algorithm, see Figure 2 and Materials and methods.

Reviewer 3 pointed out that it is unclear what is the value for the scale parameter δ that appears in the priors? Is that a free parameter? If so, do results change when this parameter varies? This seems to be a critical aspect of the process (at least insofar as the precision of the results).

We thank the reviewer for noticing this omission: δ represents the amount of partial voluming at the interface. In theory it could be tuned (in particular to be increased at smoother boundaries such as the thalamus-internal capsule interface), but setting it to the imaging scale provided the most consistent results across structures. We added this mention to the Materials and methods:

The scale parameter is set to 1 voxel, representing the expected amount of partial voluming.

[Editors' note: further revisions were suggested prior to acceptance, as described below.]

Essential revisions:Reviewer #2 (see below) would like to see a more elaborate comparison of the STN segmentation to previous recent methods, given that these sorts of subcortical parcellation methods are likely to be of interest to DBS researchers. While a comparison to the Ewert method (or similar) would be nice, a brief discussion of a subjective comparison of these results would be sufficient to address this concern.

In the article by Ewert et al., the parcellation is performed by co-registration to a very precise high resolution atlas. In their experiment (see their Table 2), they report average Cohen's Kappa of 0.4467 and 0.5240 for the STN and the GPi respectively. We recomputed the corresponding metric for our leave-one-out experiment and obtained 0.4843 and 0.5452 respectively, which is within the same range of accuracy. These structures, the STN particularly, are notoriously difficult to parcellate due to low T1 contrast with the WM of the internal capsule and the proximity of nuclei with similar T2 and T2* contrast (SN, RN GPe). However, the relatively high performance of the atlas co-registration by Ewert et al., 2018 and of multi-atlas co-registration for these structures indicates that further performance improvements may be gained by using a more anatomically precise template. We added the following to the discussion:

“Some enhancement techniques such as building a multi-subject template (Pauli et al., 2018) or adding a denoising step (Bazin et al., 2019) may be beneficial. Co-registration to a high-precision atlas as in (Ewert et al., 2018) may also improve the initial alignment over the MASSP group average template.”

and later on, when discussing applications:

“Important applications of subcortical parcellation also include deep-brain stimulation surgery (Ewert et al., 2018), where the number of structures parcellated by MASSP can help neurosurgeons orient themselves more easily, although precise targeting will still require manual refinements, especially in neurodegenerative diseases.”

Reviewer #2:[…]Additionally, I found this a little strange: Figure 3 caption: compared to the most expert of the two human raters.

We changed the text to "compared to the human rater with most neuroanatomical expertise." Here the goal was to compare to our most skilled manual delineations, rather than a consensus between expert and trainee.